# Proposal of a General Identification Method for Fractional-Order Processes Based on the Process Reaction Curve

**Juan J. Gude** [1,*] **and Pablo García Bringas** [2]

1 Department of Computing, Electronics and Communication Technologies, Faculty of Engineering, University of Deusto, 48007 Bilbao, Spain
2 Department of Mechanics, Design and Industrial Management, Faculty of Engineering, University of Deusto, 48007 Bilbao, Spain
* Correspondence: jgude@deusto.es

**Abstract:** This paper aims to present a general identification procedure for fractional first-order plus dead-time (FFOPDT) models. This identification method is general for processes having S-shaped step responses, where process information is collected from an open-loop step-test experiment, and has been conducted by fitting three arbitrary points on the process reaction curve. In order to validate this procedure and check its effectiveness for the identification of fractional-order models from the process reaction curve, analytical expressions of the FFOPDT model parameters have been obtained for both situations: as a function of any three points and three points symmetrically located on the reaction curve, respectively. Some numerical examples are provided to show the simplicity and effectiveness of the proposed procedure. Good results have been obtained in comparison with other well-recognized identification methods, especially when simplicity is emphasized. This identification procedure has also been applied to a thermal-based experimental setup in order to test its applicability and to obtain insight into the practical issues related to its implementation in a microprocessor-based control hardware. Finally, some comments and reflections about practical issues relating to industrial practice are offered in this context.

**Keywords:** process identification; fractional-order systems; fractional first-order plus dead-time model





## 1. Introduction

For the design and tuning tasks of a control system, information about the dynamic behavior of the controlled process is required, capturing it into a process mathematical model [1]. This model must provide reliable information to predict the effects that a control system will have on the behavior of the controlled process at the operating point, in particular on its output variable in servo and regulatory control.

The academic and industrial community recognizes that proportional integral derivative (PID) controller is the most widely used option in the process industry, having become an industry standard for process control [2]. Although the most commonly used models for tuning PID controllers are the first-, dual-pole, and second-order plus dead-time (FOPDT, DPPDT, SOPDT) ones, they are not able to represent the process dynamics with the required accuracy in some cases, as suggested, e.g., in [3]. Due to this, increasing the robustness of the control system can introduce issues in the well-known robustness/performance trade-off, constraining the desired performance [4]. Therefore, obtaining more accurate models of the controlled process is expected to improve performance.

In the technical literature there are a wide variety of identification procedures for integer-order models that are based on an open-loop step-test experiment; see, e.g., the identification methods detailed in [5–8] and the references cited therein. In general, these types of identification procedures are characterized by being performed with very little information about the plant, which makes them ideal for their use in industry. More specifically, some references also described identification algorithms that are based on

fitting different points on the reaction curve [9–11]. Generally, identification methods for integer-order models that are based on the location of two or three points on the process reaction curve use FOPDT, DPPDT, and SOPDT models.

Considering both two-point and three-point methods separately, in the first group, the following methods can be considered [11–14], while in the second group, methods [15–17] can be examined. The identification method proposed in [11], which is used as a two-point method for FOPDT and DPPDT models, can also be used as a three-point method for SOPDT models. In a recent paper, the method proposed in [11] has been also extended for identifying a multiple-pole with dead-time model [18].

Note that in the case of two-point methods considered above, the sets of points are asymmetrical with respect to the central point on the reaction curve (50%). The only method with a symmetrical set of points is the 123c identification method proposed by Alfaro in [11]. For the considered three-point methods, refs. [15,17] present asymmetrical sets of points, while [11] for SOPDT models and ref. [16] provide sets of points that are symmetrical, although for the latter the central point does not match the central value on the reaction curve.

In recent decades, the advent of fractional calculus and new computational techniques have made possible a major academic and industrial effort focused on the transition from classical models and controllers to those described by non-integer order differential equations. Thus, fractional-order dynamic models and controllers were introduced; see, e.g., [19,20].

The apparent benefit of fractional calculus in the field of modelling has been justified from an industrial point of view. However, it has been more difficult to convey the advantages of fractional calculus on the controller side because of implementation issues [21]. That is why the adoption of fractional-order PID controllers in the industry is currently low, even though fractional-order PID controllers offer clear advantages in comparison with integer-order ones.

Recent studies have pointed to fractional-order PID controllers as an emerging trend in process control (see [21–24]), the main reason for their success being the intrinsic robustness they offer with a higher degree of freedom to operate and tune the controller parameters. It is important to note that in some of the existing fractional-order controller design methods, a simple process model has been used in order to tune the intended controller settings [25–30].

In the technical literature there is a wide range of methods for identifying fractional-order models based on the process reaction curve; however, there are not many that are analytical techniques and whose main feature is simplicity of implementation. Some strategies to estimate the FFOPDT model parameters by using step-response data have been proposed in [31]. These combine numerical computation and graphical estimation. Integral-based estimation methods, whose main feature is their robustness to the presence of measurement noise, are proposed in [32,33].

The most common approach in industrial practice is based on nonlinear optimization; see, e.g., [34–37]. These methods are generally applied by minimizing the error between the fractional-order model step response and the process reaction curve. These techniques are characterized by the fact that they require more computational effort compared to other existing analytical methods.

Despite the fact that the fractional-order model has been demonstrated to be technologically superior on multiple occasions, the industrial adoption of the fractional-order approach requires further analysis [38].

Considering all the above, the existence of identification methods for simple-structure fractional-order models is of major relevance and can be very helpful in the practical design of integer- and fractional-order control systems. There are multiple reaction curve-based methods for identifying integer-order models, as discussed previously. This is mainly due to the fact that the step response of a process has a straightforward physical interpretation and that identification methods for integer-order models based on fitting several points of the process reaction curve are very easy to implement and apply. Therefore, one may

consider it appropriate to extend such methods for fractional models. To that end, an FFOPDT model identification method, which is based on fitting three points on the reaction curve, has been conducted in [39]. However, that study was restricted to considering that the central point is located in the center of the reaction curve (50%), and that the extreme points are located symmetrically with respect to this central point.

It is the authors' opinion that it is of significant interest to extend the identification method proposed in [39] to any set of points on the process reaction curve.

Therefore, the objective of this work is, on the one hand, to validate this identification procedure for three asymmetrical points on the reaction curve and to obtain insight into the selection of such points and their influence on the accuracy of the identified fractional-order model. On the other hand, another additional objective is to test its applicability to a laboratory prototype and to obtain some insights on the practical issues related to its implementation on a microprocessor-based control hardware.

This paper is organized as follows. Section 2 is devoted to presenting some preliminaries and theoretical background. In Section 3, a generalization of the three-point step-response method for FFOPDT models is proposed. This general method is particularized for several symmetrical and asymmetrical sets of points on the process reaction curve. The results of some numerical simulations are presented in Section 4, illustrating the effectiveness and simplicity of the proposed method for both symmetrical and asymmetrical sets of points in comparison to well-known identification methods. In Section 5, the applicability of the proposed identification procedure on a laboratory prototype is verified, showing several practical issues related to its implementation on an industrial control hardware. Finally, Section 6 presents conclusions of this paper.

## 2. Preliminaries and Theoretical Background

Some elementary definitions and basic concepts in fractional calculus are provided in this section. Elementary ideas from fractional calculus can be found in many books, such as [40,41].

The fractional integral of order $\alpha$ for function f(t) is defined as:

$$_0I_t^{\alpha}f(t) \equiv {}_0D_t^{-\alpha}f(t) = \frac{1}{\Gamma(\alpha)} \int_0^t (t-\tau)^{\alpha-1}f(\tau)d\tau \tag{1}$$

where $\Gamma(\cdot)$ is the Gamma function [40], $t \geq 0$, and $\alpha \in \mathbb{R}^+$.

The $\alpha$-th order Riemann-Liouville definition of fractional derivative of the given function f(t) is defined as:

$$_0D_t^{\alpha}f(t) = \frac{1}{\Gamma(m-\alpha)} \frac{d^m}{dt^m} \int_0^t (t-\tau)^{m-\alpha-1}f(\tau)d\tau \tag{2}$$

where $m - 1 < \alpha < m$, $m \in \mathbb{Z}^+$. The subscripts 0 and t in Definitions (1) and (2) can be considered as the limits of operation and referred to as the terminals of fractional-order integration and differentiation, respectively. For simplicity, $_0D_t^{-\alpha}$ will be denoted by $D^{-\alpha}$ and $_0D_t^{\alpha}$ by $D^{\alpha}$.

The Laplace transform of the fractional derivative based on Riemann-Liouville is:

$$L\{D^{\alpha}f(t)\} = s^{\alpha}L\{f(t)\} - \sum_{k=0}^{m-1} s^k D^{\alpha-k-1}f(0) \tag{3}$$

where $m - 1 < \alpha < m$. For zero initial conditions, (3) is reduced to:

$$L\{D^{\alpha}f(t)\} = s^{\alpha}L\{f(t)\} \tag{4}$$

A general fractional-order system can be described by a fractional differential equation of the form:

$$a_nD^{\alpha_n}y(t) + a_{n-1}D^{\alpha_{n-1}}y(t) + \cdots a_0D^{\alpha_0}y(t) = b_mD^{\beta_m}u(t) + b_{m-1}D^{\beta_{m-1}}u(t) + \cdots + b_0D^{\beta_0}u(t) \tag{5}$$

The transfer function of incommensurate real orders corresponding to the differential Equation (5) has the following expression [40]:

$$G(s) = \frac{Q(s^{\beta_k})}{P(s^{\alpha_k})} = \frac{b_ms^{\beta_m} + b_{m-1}s^{\beta_{m-1}} + \cdots + b_0s^{\beta_0}}{s^{\alpha_n} + a_{n-1}s^{\alpha_{n-1}} + \cdots + a_1s^{\alpha_1} + a_0} \tag{6}$$

where $P(s^{\alpha_k})$ and $Q(s^{\beta_k})$ have no common zeros, $a_k$ (k = 0, ... , n), $b_k$ (k = 0, ... , m) are constants, and $\alpha_k$ (k = 0, ... , n), $\beta_k$ (k = 0, ... , m) are arbitrary real or rational numbers and without loss of generality they can be arranged as $\alpha_n > \alpha_{n-1} > \cdots > \alpha_0$, and $\beta_m > \beta_{m-1} > \cdots > \beta_0$.

Considering that the transfer function G(s) given by (6) is strictly proper, then it is BIBO stable if and only if P(s) has no root in $\{\text{Re}(s) \geq 0\}$ [19].

A particular case occurs when a real number $\alpha$ exists as the greatest common divisor of $\alpha_i$, i = 1, ... , n and $\beta_i$, i = 0, ... , m. That value is referred to as the commensurate order. It holds that $\alpha_k = k\alpha$, $\beta_k = k\alpha$, $0 < \alpha < 1$, $\forall k \in \mathbb{Z}$, and the incommensurate order system (6) can also be rewritten in commensurate form as follows:

$$G(s) = \frac{Q(s^{\alpha})}{P(s^{\alpha})} = \frac{\sum_{k=0}^{m} b_k s^{k\alpha}}{\sum_{k=0}^{n} a_k s^{k\alpha}} \tag{7}$$

It has been proven that the commensurate system G(s) brought in (7) is BIBO stable if all the roots of polynomial equation P(x) = 0 in which $x = s^{\alpha}$ are positioned out of the sector $|\arg(x)| \leq \alpha\pi/2$.

Considering n > m, the function G(s) becomes a proper rational function in the complex variable $s^{\alpha}$ and, if it is supposed that roots of P(x) = 0 are distinct, the partial fraction expansion of transfer Function (7) can be written in the following general form:

$$G(s^{\alpha}) = \sum_{i=1}^{n} \frac{r_i}{s^{\alpha} + \lambda_i} \tag{8}$$

where $r_i$, i = 1, ... , n are the corresponding residues and $\lambda_i$, i = 1, ... , n are the roots of P(x) = 0.

Taking the inverse Laplace transform from (8), the impulse response of $G(s^{\alpha})$ is obtained, which is also given in [40].

$$h(t) = L^{-1}\left\{\sum_{i=1}^{n} \frac{r_i}{s^{\alpha} + \lambda_i}\right\} = \sum_{i=1}^{n} r_i t^{\alpha-1} E_{\alpha,\alpha}(-\lambda_i t^{\alpha}) \tag{9}$$

where $E_{\alpha,\alpha}(z)$ is the Mittag–Leffler function. This function is defined for an arbitrary value z as:

$$E_{\alpha,\beta}(z) = \sum_{r=0}^{\infty} \frac{z^r}{\Gamma(\alpha r + \beta)} \tag{10}$$

Integrating the right-hand side of (9), the following step response of the transfer function $G(s^{\alpha})$ is obtained:

$$g(t) = \sum_{i=1}^{n} r_i \frac{E_{\alpha,1}(-\lambda_i t^{\alpha}) - 1}{\lambda_i} \tag{11}$$

Each component of the step response g(t) in (11) converges to its final value in a similar way as function $t^{-\alpha}$ does, as has been shown in [28].

### 3. Fractional First-Order Plus Dead-Time Model Identification

The general identification procedure to be presented in this section uses process information obtained from an open-loop test and is applied only for identifying a fractional-order model for processes having an S-shaped step response.

The differential equation for an FFOPDT model can be expressed as follows:

$$T \cdot D^\alpha y(t) + y(t) = K \cdot u(t - L) \tag{12}$$

where the initial condition $y(0^+)$ is generally taken as zero to obtain a transfer function model. The controlled processes under consideration can be well characterized by this model. The standard FFOPDT model can be derived from Equation (12) by taking the Laplace transform, obtaining the following transfer function:

$$P(s) = \frac{Ke^{-Ls}}{1 + Ts^\alpha} \tag{13}$$

where K is the process gain, T > 0 is the time constant, L ≥ 0 is the apparent dead-time, and $\alpha$ is the fractional order of the model.

FFOPDT model parameters $\theta_P = \{K, T, L, \alpha\}$ can be identified using the process reaction curve, as a result of an open-loop test.

For the particular case $\alpha$ = 1, the FFOPDT model (13) becomes the standard FOPDT model, which has been broadly used in practice to capture the essential dynamic response of industrial processes for the purpose of control design [2].

The FFOPDT model (13) can be viewed as a generalization of the conventional FOPDT model, as discussed in [42], where the relevance of this generalization and the implications it has for both the identification of dynamic processes and the design of feedback control loops are indicated.

The step response of the considered fractional model can conveniently describe both monotonic or non-monotonic behaviors depending on the fractional order $\alpha$. The step responses of FFOPDT models for increasing values of $\alpha$, with $0.5 \leq \alpha \leq 1.1$ which is the range considered in this paper, are shown in Figure 1. As a particular case, the step response of the considered system for $\alpha$ = 1, which represents an FOPDT model, is depicted in red line.

In this paper, two parameters will be used to characterize process dynamics, namely $T_{ar}$ and $\tau$, which are two classical parameters well-defined for integer-order processes and can also be defined for the fractional case.

$T_{ar}$ is referred to as the average residence time and has been defined in the fractional context as follows [39]:

$$T_{ar} \doteq \frac{\int_0^\infty tg(t)dt}{\int_0^\infty g(t)dt} = L + T^{1/\alpha} \tag{14}$$

where g(t) represents the impulse response of the fractional-order process. The average residence time is a classical index that characterizes process dynamics by indicating the time it takes the input to have a significant influence on the output [2].

$\tau$ is referred to as the normalized dead-time, the typical range of which is $0 \leq \tau \leq 1$, and has been defined in the fractional context as follows [39]:

$$\tau \doteq \frac{L}{T_{ar}} = \frac{L}{L + T^{1/\alpha}} \tag{15}$$

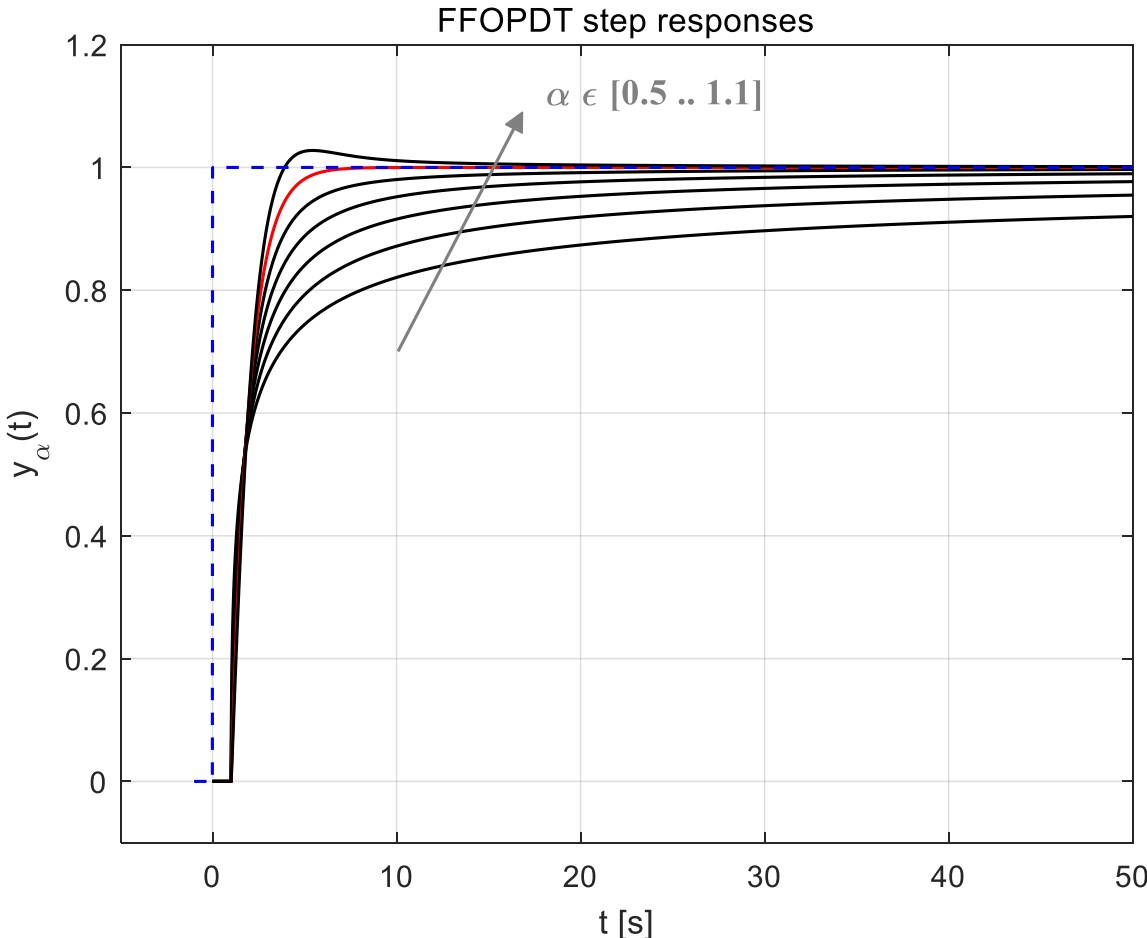

**Figure 1.** Step responses of FFOPDT models with $\alpha \in [0.5, 1.1]$. The step response of an FOPDT model ($\alpha = 1$) is depicted in red line.

This parameter has been defined to characterize the degree of difficulty in controlling a process. Broadly speaking, processes with small $\tau$ can be considered to be easy to control, while processes with a larger value of $\tau$ are difficult to control. The standard definition of the classical parameters $T_{ar}$ and $\tau$ for the case $\alpha = 1$ corresponds to the one for an FOPDT model [2].

Although with an FFOPDT model the monotonic and non-monotonic behavior of the process can be characterized, in this work we will only focus on processes with monotonic response with $0.50 \leq \alpha \leq 1.00$, characterized by an S-shaped response. The importance of processes with essentially monotone step responses lies in the fact that they are very common in process control [2].

The FFOPDT model normalized step responses for different values of $\alpha$ and different values of $\tau$ are shown in Figure 2. It can be noticed from the figure that all responses have a common point at $t = T_{ar}$ because time is normalized with respect to the average residence time.

Considering that a step signal u(t) with an amplitude $\Delta u$ is applied to an FFOPDT model (13), a signal $y_\alpha(t)$ with an amplitude variation of $\Delta y$ is obtained, as shown in Figure 3.

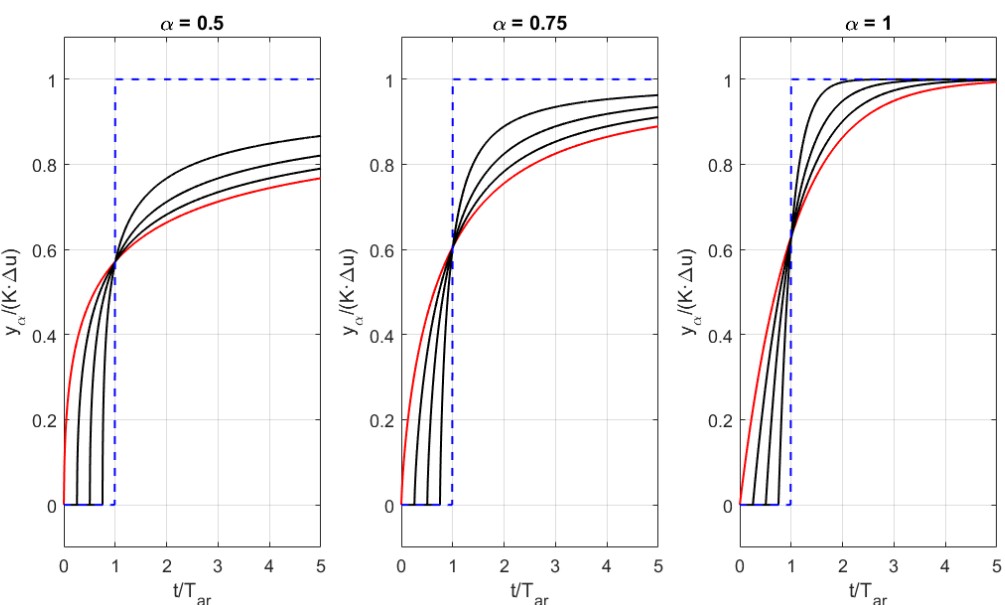

**Figure 2.** Normalized step responses $y_\alpha/(K \cdot \Delta u)$ vs. normalized time, where FFOPDT models for $\alpha = 0.5$, 0.75, and 1.0 and $\tau = 0$ (red), 0.25, 0.5, 0.75, and 1 (blue) have been considered. Note that all step responses have a common point at $t = T_{ar}$.

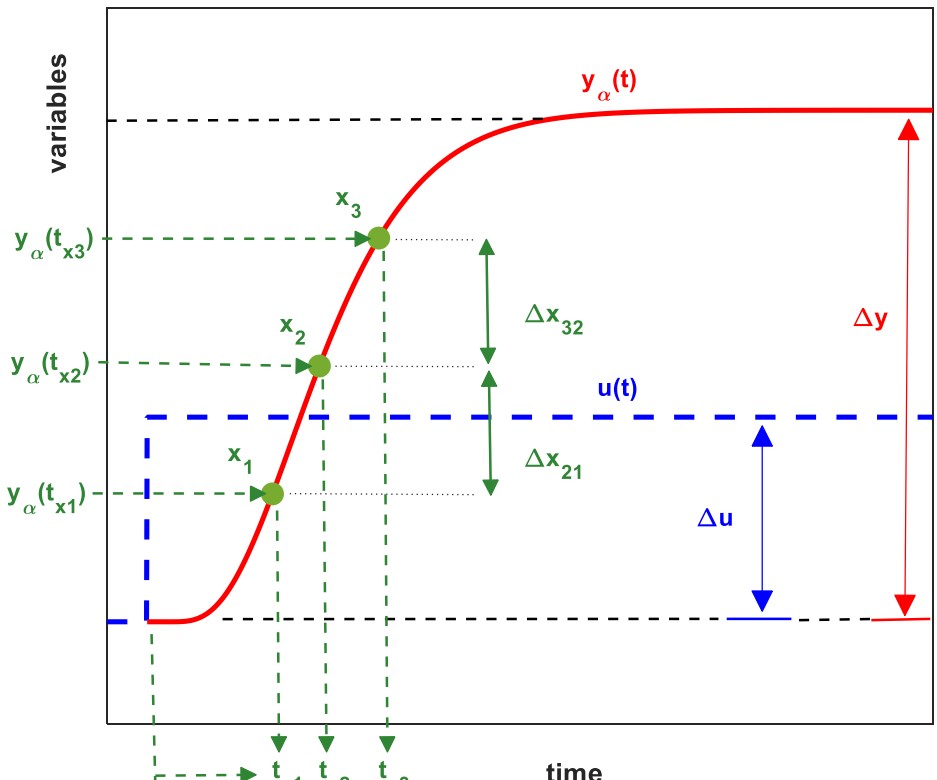

**Figure 3.** Process reaction curve $y_\alpha(t)$, arbitrary representative points $\{x_1, x_2, \text{and } x_3\}$ on the process reaction curve, and step-input signal $u(t)$. Data obtained from the process reaction curve required for the identification procedure are $\{\Delta y, \Delta u, t_{x1}, t_{x2}, t_{x3}\}$. Note that $\Delta x_{21}$ and $\Delta x_{32}$ represent a variation of $(x_2 - x_1)\%$ and $(x_3 - x_2)\%$ from the centroid $x_2$ to the extreme points $x_1$ and $x_3$, respectively.

The corresponding FFOPDT model response to a $\Delta u$ step input change is:

$$y_\alpha(t) = \begin{cases} 0, & 0 \le t < L \\ K\left\{1 - E_{\alpha,1}\left[-\frac{1}{T}(t-L)^\alpha\right]\right\}\Delta u, & t \ge L \end{cases} \tag{16}$$

where $E_{\alpha,\beta}$ is the two-parameter Mittag–Leffler function, which was previously defined in (10).

Normalizing the process output $y_\alpha(t)$ with respect to its final value $\Delta y = K \cdot \Delta u$ and using the shifted and normalized time $\tau = \frac{1}{T}(t-L)^\alpha$, Equation (16) is reduced to:

$$\widetilde{y}_\alpha(\tau) = 1 - E_{\alpha,1}(-\tau), \quad \tau \ge 0 \tag{17}$$

The FFOPDT model parameters, $\theta_P = \{K, T, L, \alpha\}$, will be determined considering the step response of an FFOPDT model (16) and the normalized process output (17), respectively.

From Equation (16), the gain is given by:

$$K = \frac{\Delta y}{\Delta u} \tag{18}$$

where $\Delta y$ is the total process output change when a step input with amplitude $\Delta u$ is applied, as indicated in Figure 3.

Since $\widetilde{y}_\alpha(\tau)$ is a specific value of the normalized output, which has the following range $0 \le \widetilde{y}_\alpha(\tau) \le 1$, its corresponding normalized time $\tau_x$ can be found using (17). Therefore, the time $t_x$ required for the process output (16) to reach such x-point is:

$$t_x = L + (\tau_x T)^{1/\alpha} \tag{19}$$

In order to obtain the rest of the FFOPDT model parameters ($T$, $L$, and $\alpha$), the set of times $\{t_{x1}, t_{x2}, t_{x3}\}$ to reach points $\{y_\alpha(t_{x1}), y_\alpha(t_{x2}), y_\alpha(t_{x3})\}$, respectively, on the process reaction curve are required.

Note that, without loss of generality, $t_{x1} < t_{x2} < t_{x3}$ and $\tau_{x1} < \tau_{x2} < \tau_{x3}$ will be considered in this paper.

Considering expressions like (19), the following equations set is defined:

$$\begin{aligned} t_{x1} &= L + (\tau_{x1}T)^{1/\alpha} \\ t_{x2} &= L + (\tau_{x2}T)^{1/\alpha} \\ t_{x3} &= L + (\tau_{x3}T)^{1/\alpha} \end{aligned} \tag{20}$$

In order for the model fractional order $\alpha$ to be estimated, the following ratio index $\Delta$ is found by considering the previously obtained set of Equation (20):

$$\Delta \doteq \frac{t_{x3} - t_{x1}}{t_{x2} - t_{x1}} = \frac{\tau_{x3}^{1/\alpha} - \tau_{x1}^{1/\alpha}}{\tau_{x2}^{1/\alpha} - \tau_{x1}^{1/\alpha}} \tag{21}$$

where $\tau_{x1}$, $\tau_{x2}$, and $\tau_{x3}$ are normalized times defined by (19) and which can be determined using Equation (17). Note that $\Delta$ index can be interpreted as the ratio between the time difference in reaching from $x_1$ to $x_3$% and from $x_1$ to $x_2$% of the total variation of the process output, as indicated in (21). From Equation (21) it is clear that there is a dependence between fractional order $\alpha$ and ratio index $\Delta$, which can be expressed as $\alpha = f_1(\Delta)$.

For obtaining the time-based parameters ($T$, $L$) from (20), it is required to consider the following two points, $\{t_{x1}, y_\alpha(t_{x1})\}$ and $\{t_{x3}, y_\alpha(t_{x3})\}$, on the process reaction curve. Then, the expressions for these parameters are:

$$T = a^\alpha (t_{x3} - t_{x1})^\alpha \tag{22}$$

and

$$L = t_{x3} - \tau_{x3}^{1/\alpha}T^{1/\alpha} \tag{23}$$

where

$$a = \frac{1}{\tau_{x3}^{1/\alpha} - \tau_{x1}^{1/\alpha}} \tag{24}$$

The two equivalent normalized points for obtaining parameters T and L are $\{\tilde{y}_\alpha(\tau_{x1}), \tau_{x1}\}$ and $\{\tilde{y}_\alpha(\tau_{x3}), \tau_{x3}\}$.

The set of Equation (25) are the expressions required to obtain the FFOPDT model parameters, $\theta_P = \{K, T, L, \alpha\}$, by using the times needed for the response to reach any three points on the process reaction curve $\{x_1\text{-}x_2\text{-}x_3\%\}$.

$$\begin{cases} K = \frac{\Delta y}{\Delta u} \\ \alpha = f_1(\Delta) \\ T = f_2(\alpha)(t_{x3} - t_{x1})^\alpha \\ L = \max\left[t_{x3} - f_3(\alpha)T^{1/\alpha},\ 0\right] \end{cases} \tag{25}$$

where $\Delta y$ is the total process output change to the step input $\Delta u$, as shown in Figure 3; function $f_1$ depends on $\Delta$ and is defined in (21) as a function of the times $\{t_{x1}, t_{x2}, t_{x3}\}$; functions $f_2(\alpha) = a^\alpha$ and $f_3(\alpha) = \tau_{x3}^{1/\alpha}$ depend on $\tau_{x1}$ and $\tau_{x3}$, and $\tau_{x3}$, respectively, and finally a-parameter is defined in (24). It can be observed from (25) that the expressions of T and L have a high dependence on the value of $\alpha$. This makes it of significant importance to determine the value of $\alpha$ parameter accurately.

In (25), $\alpha > 0$ and $T > 0$ are fulfilled in a natural way, since $\tau_{x1} < \tau_{x2} < \tau_{x3}$ and $t_{x1} < t_{x2} < t_{x3}$. The following condition must be satisfied to ensure $L \geq 0$:

$$t_{x3} \geq \tau_{x3}^{1/\alpha} \tag{26}$$

For a more detailed development of Equation (25) we refer the reader to [39], where these equations are obtained and subsequently particularized for the case in which the three points on the process reaction curve are symmetrical with respect to the central point.

Figure 4 shows the general scheme of the complete procedure for obtaining the expressions for the identification of the FFOPDT model parameters, $\theta_P = \{K, T, L, \alpha\}$, from three arbitrary points of the process reaction curve.

This procedure is summarized in the following steps, as depicted in Figure 4.

1.  From the normalized process output (17), $\tilde{y}_\alpha(\tau)$, the values of the normalized times $\{\tau_{x1}, \tau_{x2}, \tau_{x3}\}$ of the three considered points on the process reaction curve are obtained for the different values of $\alpha$, $0.50 \leq \alpha \leq 1.10$.
2.  Data sets $\{\Delta, \alpha\}$, $\{\alpha, a\alpha\}$, and $\{\alpha, (\tau x3)1/\alpha\}$ are obtained for the considered set of points $(x_1\text{-}x_2\text{-}x_3\%)$ by using the values of the corresponding normalized times.

Note that this procedure is general and admits the points $x_1$, $x_2$, and $x_3$ to be arbitrary. In this paper, both a symmetrical and an asymmetrical location of the points on the process reaction curve will be considered.

3.  By means of a curve-fitting procedure, the values of the parameters $\{pi, qi\}$ for the rational functions $\alpha = f1(\Delta)$, $f2(\alpha)$, and $f3(\alpha)$, respectively, are obtained.
4.  From the rational functions $\alpha = f1(\Delta)$, $f2(\alpha)$, and $f3(\alpha)$ obtained in the previous step, the expressions for the FFOPDT model parameters (25) are completed.
5.  Once the numerical values of $\alpha$, $f2$, and $f3$ are determined, the values of the FFOPDT model parameters, $\theta_P = \{K, T, L, \alpha\}$, are calculated using expressions (25) and experimental data collected from the process reaction curve, $\{\Delta y, \Delta u, t_{x1}, t_{x2}, t_{x3}\}$.

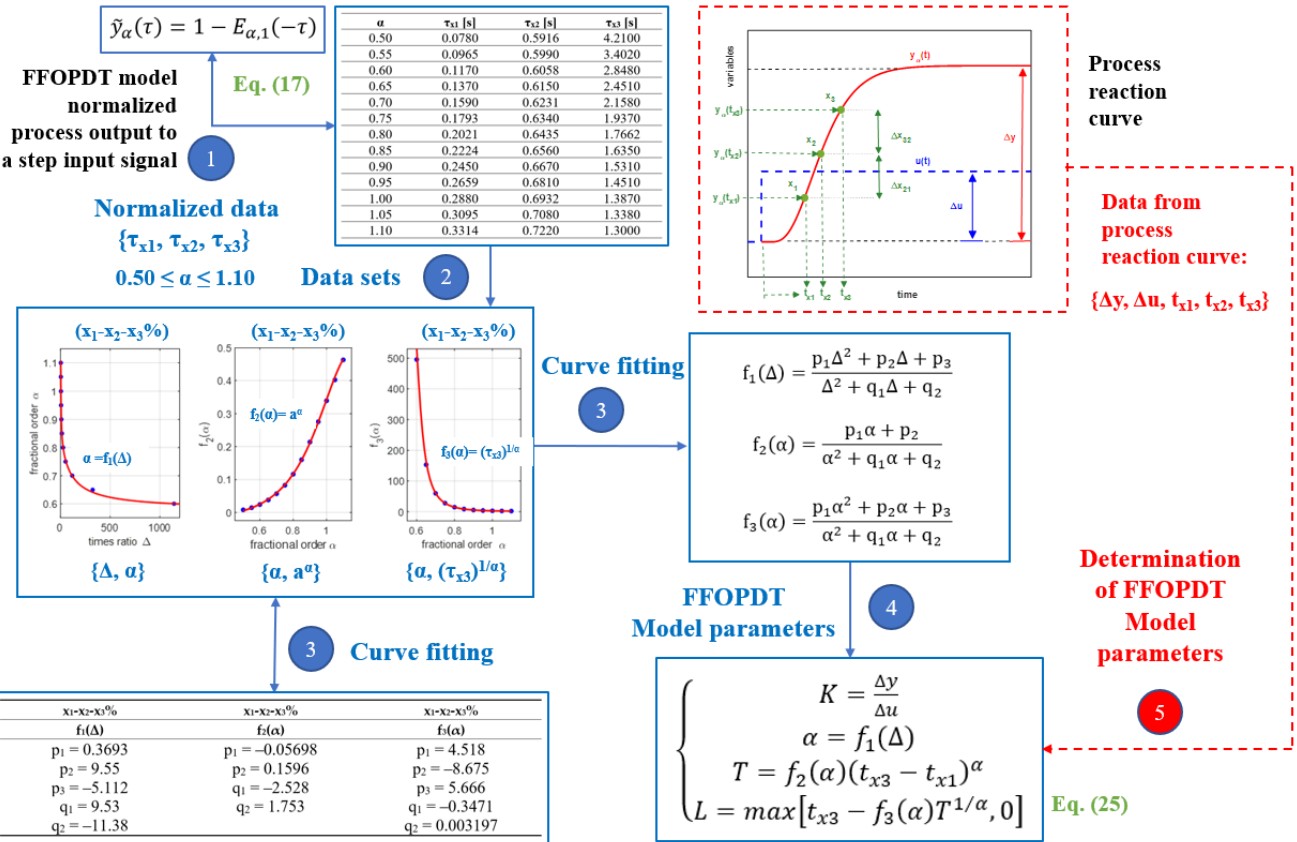

**Figure 4.** Scheme of the complete procedure for identifying the parameters of the fractional-order model considering three arbitrary points on the process reaction curve. Note that the blue part of the scheme (steps 1–4) represents the general procedure to obtain expressions (25) that allow to determine the parameters of the FFOPDT model for any three points ($x_1$, $x_2$, and $x_3$) on the process reaction curve. On the other hand, the red part of the scheme (step 5) indicates how to estimate the parameters $\theta_P = \{K, T, L, \alpha\}$ from the information collected from the process reaction curve $\{\Delta y, \Delta u, t_{x1}, t_{x2}, t_{x3}\}$.

A detailed algorithm for estimating the FFOPDT model parameters, $\theta_P = \{K, T, L, \alpha\}$, from the information collected from the process reaction curve is provided in Section 5.4. This algorithm, which simplifies its software implementation, can serve as a complement to the scheme represented in Figure 4.

Table 1 includes the numerical values of the normalized times $\tau_5$, $\tau_{10}$, $\tau_{20}$, $\tau_{25}$, $\tau_{50}$, $\tau_{55}$, $\tau_{60}$, $\tau_{75}$, $\tau_{90}$, and $\tau_{95}$, respectively. These data will be used indistinctly as $\tau_x$, $\tau_{50}$, or $\tau_{100-x}$ in Section 3.1 for a symmetrical set of points (x-50-(100 − x)%), and $\tau_{x1}$, $\tau_{x2}$, or $\tau_{x3}$ in Section 3.2 for an asymmetrical set of points ($x_1$-$x_2$-$x_3$%), respectively, and constitute the main source of data to determine the corresponding data sets $\{\Delta, \alpha\}$, $\{\alpha, a^\alpha\}$, and $\{\alpha, (\tau_{x3})^{1/\alpha}\}$ or $\{\alpha, (\tau_{100-x})^{1/\alpha}\}$, needed to determine expressions for the functions $\alpha = f_1(\Delta)$, $f_2(\alpha)$, and $f_3(\alpha)$.

### 3.1. Symmetrical Set of Points (x-50-(100 − x)%)

This case can be considered as a simplification of the general identification procedure, where only points that are symmetrically located on the process reaction curve are selected. Note that the central point or centroid will be located in the middle of the range, $x_2 = 50\%$, ($t_{x2} = t_{50}$, $y_\alpha(t_{50})$), as depicted in Figure 3. The remaining two points could be located arbitrarily on the reaction curve, but symmetrically located with respect to the central point ($\Delta x_{32} = \Delta x_{21}$). One of the extreme points will be denoted $x_1 = x$, the other being $x_3 = 100$ − x. In this case, the times to be determined will be $t_{x1} = t_x$ and $t_{x3} = t_{100-x}$, where $t_x$ and

$t_{100-x}$ denote the time required to reach x% ($y_\alpha(t_x)$) and $(100-x)$% ($y_\alpha(t_{100-x})$) of the total process output change, respectively, considering the following range $0 < x < 50$.

**Table 1.** Numeric values of normalized times $\tau_5$, $\tau_{10}$, $\tau_{20}$, $\tau_{25}$, $\tau_{50}$, $\tau_{55}$, $\tau_{60}$, $\tau_{75}$, $\tau_{90}$, and $\tau_{95}$, respectively, for different values of $\alpha$, with $0.5 \leq \alpha \leq 1.1$.

| $\alpha$ | $\tau_5$ [s] | $\tau_{10}$ [s] | $\tau_{20}$ [s] | $\tau_{25}$ [s] | $\tau_{50}$ [s] | $\tau_{55}$ [s] | $\tau_{60}$ [s] | $\tau_{75}$ [s] | $\tau_{90}$ [s] | $\tau_{95}$ [s] |
|---|---|---|---|---|---|---|---|---|---|---|
| 0.5 | 0.0462 | 0.0963 | 0.2113 | 0.2781 | 0.7691 | 0.9216 | 1.1072 | 2.0516 | 5.5556 | 11.3640 |
| 0.6 | 0.0464 | 0.0965 | 0.2101 | 0.2752 | 0.7402 | 0.8810 | 1.0481 | 1.8734 | 4.7612 | 9.3440 |
| 0.7 | 0.0471 | 0.0975 | 0.2107 | 0.2749 | 0.7181 | 0.8470 | 0.9988 | 1.7127 | 3.9916 | 7.4160 |
| 0.8 | 0.0481 | 0.0993 | 0.2131 | 0.2768 | 0.7028 | 0.8220 | 0.9601 | 1.5757 | 3.2873 | 5.5890 |
| 0.9 | 0.0495 | 0.1019 | 0.2172 | 0.2811 | 0.6945 | 0.8060 | 0.9326 | 1.4665 | 2.7112 | 4.0190 |
| 1.0 | 0.0513 | 0.1054 | 0.2232 | 0.2877 | 0.6932 | 0.7990 | 0.9163 | 1.3863 | 2.3026 | 2.9960 |
| 1.1 | 0.0536 | 0.1097 | 0.2309 | 0.2967 | 0.6988 | 0.8000 | 0.9109 | 1.3334 | 2.0419 | 2.4560 |

In this section, the sets of points indicated in Table 2 will be considered. For the case of symmetrical points, sets #1 and #2 have been chosen because the accuracy of the fractional-order model is improved for low values of x. In particular, the influence of the location of the symmetrical representative points of the reaction curve on the accuracy of the identified model is explained in detail in [39]. In addition, set #3 has been chosen because it gives very good results for integer- or close to integer-order models, see also [39]. This behavior has already been observed in the technical literature in some identification methods for integer-order models, e.g., the aforementioned 123c method [11] uses time sets (25–75%) to identify FOPDT and DPPDT models and (25-50-75%) for SOPDT models; and in [18], points (25-50-75%) are proposed to identify multiple-pole with dead-time models.

**Table 2.** Sets of symmetrical points that have been considered.

| Set # | Symmetrical Points | Centroid | Distance from Centroid |
|---|---|---|---|
| 1 | (5-50-95%) | $x_2 = 50$% | $\Delta x_{21} = \Delta x_{32} = 45$% |
| 2 | (10-50-90%) | $x_2 = 50$% | $\Delta x_{21} = \Delta x_{32} = 40$% |
| 3 | (25-50-75%) | $x_2 = 50$% | $\Delta x_{21} = \Delta x_{32} = 25$% |

Then, the procedure summarized in Figure 4 is followed for the symmetrical case. Data sets $\{\Delta, \alpha\}$, $\{\alpha, a^\alpha\}$, and $\{\alpha, (\tau_{100-x})^{1/\alpha}\}$ for $0.5 \leq \alpha \leq 1.1$, and the functions $f_1(\Delta)$, $f_2(\alpha)$, and $f_3(\alpha)$ obtained by curve fitting for the different sets of symmetrical points, respectively, are shown in Figure 5. Note that $\Delta$ depends on normalized times $\{\tau_x, \tau_{50}, \text{ and } \tau_{100-x}\}$ and has a significant dependence on $\alpha$ parameter, that $f_2(\alpha) = a^\alpha$ depends on $\alpha$ and normalized times $\tau_x$ and $\tau_{100-x}$, and that $f_3(\alpha) = (\tau_{100-x})^{1/\alpha}$ depends on $\alpha$ and normalized time $\tau_{100-x}$.

The following rational functions have been used for curve fitting of functions $f_1(\Delta)$, $f_2(\alpha)$, and $f_3(\alpha)$, respectively:

$$f_1(\Delta) = \frac{p_1\Delta^2 + p_2\Delta + p_3}{\Delta^2 + q_1\Delta + q_2} \tag{27}$$

$$f_2(\alpha) = \frac{p_1\alpha + p_2}{\alpha^2 + q_1\alpha + q_2} \tag{28}$$

$$f_3(\alpha) = \frac{p_1\alpha^2 + p_2\alpha + p_3}{\alpha^2 + q_1\alpha + q_2} \tag{29}$$

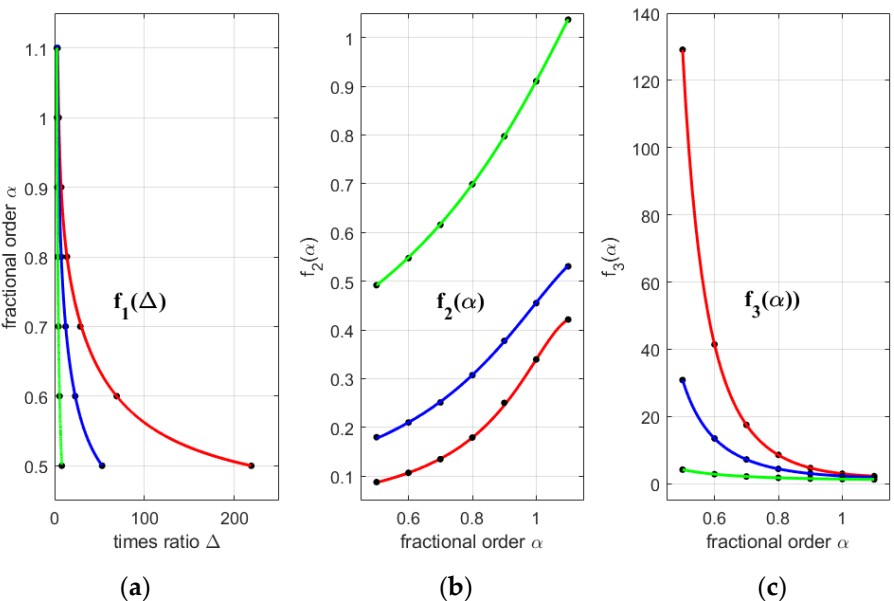

**Figure 5.** Data sets and results of curve fitting for symmetrical sets of points, (5-50-95%), (10-50-90%), and (25-50-75%), respectively: (**a**) Data sets $\{\Delta, \alpha\}$ and curve fitting for $f_1(\Delta)$; (**b**) Data sets $\{\alpha, a^\alpha\}$ and curve fitting for $f_2(\alpha)$; (**c**) Data sets $\{\alpha, (\tau_{100-x})^{1/\alpha}\}$ and curve fitting for $f_3(\alpha)$. Note that in each graph, curve fitting for (5-50-95%) is represented in red, (10-50-90%) in blue, and (25-50-75%) in green.

The Levenberg–Marquardt least-squares curve-fitting algorithm has been used for fitting data in all graphs in Figure 5. The values of the corresponding parameters $\{p_i, q_i\}$ for functions $f_1(\Delta)$, $f_2(\alpha)$, and $f_3(\alpha)$, and each one of the selected sets of symmetrical points are shown in Tables 3–5, respectively.

### 3.2. Asymmetrical Set of Points ($x_1$-$x_2$-$x_3$%)

In this section, the general identification procedure is applied to three arbitrary points asymmetrically located on the reaction curve ($x_1$, $x_2$, and $x_3$), as shown in Figure 3. This means that the times to be determined will be $t_{x1}$, $t_{x2}$, and $t_{x3}$, where they denote the time needed to reach $x_1$% ($y_\alpha(t_{x1})$), $x_2$% ($y_\alpha(t_{x2})$), and $x_3$% ($y_\alpha(t_{x3})$) of the total process output change, respectively.

**Table 3.** Parameters $\{p_i, q_i\}$ of the rational function $f_1(\Delta)$ for the different sets of symmetrical points.

| (5-50-95%) | (10-50-90%) | (25-50-75%) |
|---|---|---|
| $p_1 = 0.4259$ | $p_1 = 0.3808$ | $p_1 = 0.2676$ |
| $p_2 = 38.78$ | $p_2 = 13.57$ | $p_2 = 1.756$ |
| $p_3 = 14.34$ | $p_3 = -3.067$ | $p_3 = -2.578$ |
| $q_1 = 45.33$ | $q_1 = 14.69$ | $q_1 = -0.7042$ |
| $q_2 = -27.8$ | $q_2 = -15.9$ | $q_2 = -1.289$ |

**Table 4.** Parameters $\{p_i, q_i\}$ of the rational function $f2(\alpha)$ for the different sets of symmetrical points.

| (5-50-95%) | (10-50-90%) | (25-50-75%) |
|---|---|---|
| $p_1 = -0.0337$ | $p_1 = -0.05698$ | $p_1 = -0.3443$ |
| $p_2 = 0.0595$ | $p_2 = 0.1596$ | $p_2 = 0.7806$ |
| $q_1 = -2.328$ | $q_1 = -2.528$ | $q_1 = -3.017$ |
| $q_2 = 1.404$ | $q_2 = 1.753$ | $q_2 = 2.496$ |

**Table 5.** Parameters $\{p_i, q_i\}$ of the rational function $f_3(\alpha)$ for the different sets of symmetrical points.

| (5-50-95%) | (10-50-90%) | (25-50-75%) |
|---|---|---|
| $p_1 = 10.43$ | $p_1 = 4.518$ | $p_1 = 3.901$ |
| $p_2 = -22.09$ | $p_2 = -8.675$ | $p_2 = -4.833$ |
| $p_3 = 13.14$ | $p_3 = 5.666$ | $p_3 = 4.546$ |
| $q_1 = -0.586$ | $q_1 = -0.3471$ | $q_1 = 2.239$ |
| $q_2 = 0.07943$ | $q_2 = 0.003197$ | $q_2 = -0.6319$ |

The origin of the asymmetry comes not only from the fact that, in general, the centroid $x_2$ will not be located in the middle of the range (0–100%) of the process output, but also from the fact that the distance from the centroid $x_2$ to the extreme points $x_1$ and $x_3$, respectively, is different, i.e., $\Delta x_{21} \neq \Delta x_{32}$.

As discussed previously, the objective is to validate this identification procedure for the asymmetrical case and to obtain insight into the selection of representative points of the reaction curve and their influence on the accuracy of the identified model.

In this section, the sets of points indicated in Table 6 will be considered.

**Table 6.** Sets of asymmetrical points that have been considered.

| Set # | Asymmetrical Points | Centroid | Distance from Centroid |
|---|---|---|---|
| 4 | (10-55-90%) | $x_2 = 55\%$ | $\Delta x_{21} = 45\%, \Delta x_{32} = 35\%$ |
| 5 | (20-60-95%) | $x_2 = 60\%$ | $\Delta x_{21} = 40\%, \Delta x_{32} = 35\%$ |
| 6 | (20-75-95%) | $x_2 = 75\%$ | $\Delta x_{21} = 55\%, \Delta x_{32} = 25\%$ |

The selection of these three sets of points ($x_1$-$x_2$-$x_3$%) is based on experimentation. A large number of experiments have been performed in order to draw the following observations:

1.  The three sets of points have been chosen with a high $x_3$ value, where $x_3 = 90$ or 95%, because the obtained model fits better the reaction curve, especially in the final part. In this regard, it has been shown in [39] that the step response of the identified models gives a good fit with the process reaction curve for the symmetrical case, particularly in the interval $[x$-$(100 - x)]$. Due to the symmetry exhibited by this method, the interval $[x$-$(100 - x)]$ is larger for lower values of x and, therefore, the step response of these models fits better the process reaction curve, which translates into a lower value in the performance index S for this fractional-order model.

2.  In general, the selection of $x_1$ affects the accuracy of the model in the initial part and, together with $x_3$, allows better fitting of T parameter.

3.  With respect to the centroid $x_2$, set #4 has been chosen in order to test the effect of moving the centroid $x_2$, increasing $\Delta x_{21}$ and decreasing $\Delta x_{32}$, in comparison with the symmetrical set #2. Sets #5 and #6 have been chosen because the effect of moving the centroid $x_2$, while keeping the extreme values $x_1$ and $x_3$, can be observed. In particular, set #5 allows us to analyze the effect of increasing $x_1$, with asymmetric distances $\Delta x_{21} = 40\%$ and $\Delta x_{32} = 35\%$, while set #6 shows the effect of increasing $x_2$, with asymmetric distances $\Delta x_{21} = 55\%$ and $\Delta x_{32} = 25\%$.

Although the sets of points could have been chosen considering other criteria, in the authors' opinion, these are the ones that best reflect the effect of their position on the model's accuracy.

In the following section, several examples will be used to verify that these observations are met.

Then, the procedure summarized in Figure 4 is followed for the asymmetrical case. Data sets $\{\Delta, \alpha\}$, $\{\alpha, a^\alpha\}$, and $\{\alpha, (\tau_{100-x})^{1/\alpha}\}$ for $0.5 \leq \alpha \leq 1.1$, and the functions $f_1(\Delta)$, $f_2(\alpha)$, and $f_3(\alpha)$ obtained by curve fitting for the different sets of asymmetrical points, i.e., (10-55-90%), (20-60-95%), and (20-75-95%), respectively, are shown in Figure 6. Note that

$\Delta$ depends on normalized times $\{\tau_{x1}, \tau_{x2}, \text{ and } \tau_{x3}\}$ and has a significant dependence on $\alpha$, that $f_2(\alpha) = a^\alpha$ depends on $\alpha$ and normalized times $\tau_{x1}$ and $\tau_{x3}$, and that $f_3(\alpha) = (\tau_{x3})^{1/\alpha}$ depends on $\alpha$ and normalized time $\tau_{100-x}$.

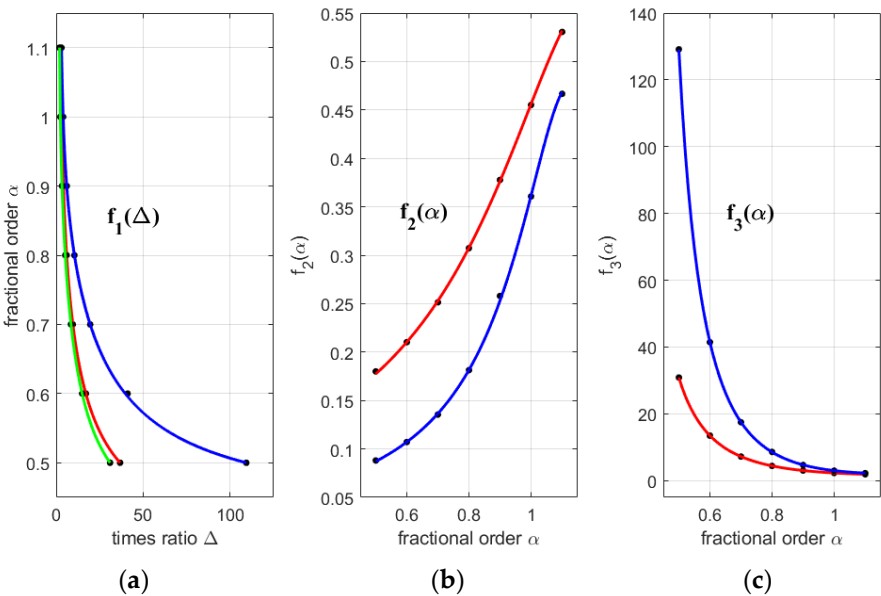

| (a) | (b) | (c) |

**Figure 6.** Data sets and results of curve fitting for asymmetrical sets of points, (10-55-90%), (20-60-95%), and (20-75-95%), respectively: (**a**) Data sets $\{\Delta, \alpha\}$ and curve fitting for $f_1(\Delta)$; (**b**) Data sets $\{\alpha, a^\alpha\}$ and curve fitting for $f_2(\alpha)$; (**c**) Data sets $\{\alpha, (\tau_{x3})^{1/\alpha}\}$ and curve fitting for $f_3(\alpha)$. Note that in each graph, curve fitting for (10-55-90%) is represented in red, (20-60-95%) in blue, and (20-75-95%) in green, respectively. Note also that sets #5 (20-60-95%) and #6 (20-75-95%) have the same functions $f_2(\alpha)$ and $f_3(\alpha)$.

The Levenberg–Marquardt least-squares curve-fitting algorithm has been used for fitting data in all graphs in Figure 6. The same rational functions (27)–(29) have been used for obtaining functions $f_1(\Delta)$, $f_2(\alpha)$, and $f_3(\alpha)$, respectively, by using curve fitting.

The values of the corresponding parameters $\{p_i, q_i\}$ for functions $f_1(\Delta)$, $f_2(\alpha)$, and $f_3(\alpha)$, and each one of the selected sets of asymmetrical points are shown in Tables 7–9, respectively.

**Table 7.** Parameters $\{p_i, q_i\}$ of the rational function $f_1(\Delta)$ for the different sets of asymmetrical points.

| (10-55-90%) | (20-60-95%) | (20-75-95%) |
|---|---|---|
| $p_1 = 0.3693$ | $p_1 = 0.4165$ | $p_1 = 0.3665$ |
| $p_2 = 9.55$ | $p_2 = 18.5$ | $p_2 = 7.191$ |
| $p_3 = -5.112$ | $p_3 = -15.5$ | $p_3 = -8.393$ |
| $q_1 = 9.53$ | $q_1 = 18.69$ | $q_1 = 5.838$ |
| $q_2 = -11.38$ | $q_2 = -25.6$ | $q_2 = -8.767$ |

**Table 8.** Parameters $\{p_i, q_i\}$ of the rational function $f_2(\alpha)$ for the different sets of asymmetrical points. Note that the parameters for (20-60-95%) and (20-75-95%) are the same since their $x_1$ and $x_3$-values are the same ($x_1 = 20\%$ and $x_3 = 95\%$).

| (10-55-90%) | (20-60-95%) | (20-75-95%) |
|---|---|---|
| $p_1 = -0.05698$ | $p_1 = -0.03498$ | $p_1 = -0.03498$ |
| $p_2 = 0.1596$ | $p_2 = 0.05957$ | $p_2 = 0.05957$ |
| $q_1 = -2.528$ | $q_1 = -2.33$ | $q_1 = -2.33$ |
| $q_2 = 1.753$ | $q_2 = 1.398$ | $q_2 = 1.398$ |

**Table 9.** Parameters $\{p_i, q_i\}$ of the rational function $f_3(\alpha)$ for the different sets of asymmetrical points. Note that the parameters for (20-60-95%) and (20-75-95%) are the same since their $x_3$-values are the same ($x_3 = 95$%).

| (10-55-90%) | (20-60-95%) | (20-75-95%) |
|:---:|:---:|:---:|
| $p_1 = 4.518$ | $p_1 = 10.43$ | $p_1 = 10.43$ |
| $p_2 = -8.675$ | $p_2 = -22.09$ | $p_2 = -22.09$ |
| $p_3 = 5.666$ | $p_3 = 13.14$ | $p_3 = 13.14$ |
| $q_1 = -0.3471$ | $q_1 = -0.586$ | $q_1 = -0.586$ |
| $q_2 = 0.003197$ | $q_2 = 0.07943$ | $q_2 = 0.07943$ |

Note that the obtained values of T, L, and $\alpha$, which are determined by using Equations (25), depend on functions $f_1(\Delta)$, $f_2(\alpha)$, and $f_3(\alpha)$. These functions thus play a relevant role in the identification method as the features of normalized step responses (17) can be well characterized due to their respective contribution. It is important to emphasize that, for any different choice of times set $\{t_{x1}\ t_{x2},\ t_{x3}\}$ to reach three points $\{y_\alpha(t_{x1}),\ y_\alpha(t_{x2}),\ y_\alpha(t_{x3})\}$ on the reaction curve, the accuracy of the identification results only depends upon the fitting precision. In this context, an accurate determination of $\alpha$-value is of primary importance since, subsequently, functions $f_2$ and $f_3$—and therefore T and L—depend on the estimated value of $\alpha$.

## 4. Simulation Results

In this section, the identification method proposed in this work has been proved for several models that exhibit fractional behavior.

Process models (30), (31), and (32) have been selected to test the effectiveness of the proposed method in obtaining an FFOPDT model (13) in comparison with several identification methods for integer- and fractional-order models.

$$P_1(s) = \frac{1}{(1 + s^{0.75})^2} e^{-0.1s} \tag{30}$$

$$P_2(s) = \frac{3}{(1 + 3s^{0.88})(1 + 2s^{0.88})(1 + s^{0.88})} \tag{31}$$

$$P_3(s) = e^{-\sqrt{s}} \tag{32}$$

On the one hand, process $P_1$ has been used to evaluate the proposed identification method for several symmetrical and asymmetrical sets of points in comparison with several well-known identification methods for integer-order models and to obtain insight into the influence of the location of asymmetrical points on the accuracy of the identified model.

On the other hand, processes $P_2$ and $P_3$ have been used to evaluate the model performance of the proposed procedure for symmetrical and asymmetrical points with other fractional-order methods.

These examples selected in this section have been also used to validate the proposed identification method for both symmetrical and asymmetrical points on the process reaction curve.

The experimental procedure followed is as follows: A step signal has been applied to the input of these processes and the reaction curve has been registered. Then the process output responses have been used to obtain the parameters of the models using the different identification methods. The sampling period used in all the experiments is Ts = 10 ms.

Finally, it is necessary to evaluate the accuracy of the model parameters that have been identified and the effectiveness of the model structure that has been adopted. A wide variety of model validation methods and fitting objective functions for system identification are available in the technical literature [5].

Although another objective function could have been used, the Mean Squared Error (MSE) has been used as a time-domain fitting criterion as a measure of performance for the identified model:

$$S(\overline{\theta}) = \frac{1}{N_s} \sum_{k=1}^{N_s} \left[ e(kT_s, \overline{\theta}) \right]^2 = \frac{1}{N_s} \sum_{k=1}^{N_s} \left[ y(kT_s) - y_m(kT_s, \overline{\theta}) \right]^2 \tag{33}$$

where $e(kT_s, \overline{\theta})$ is the difference between the process reaction curve and the step response of the identified model, $y(kT_s)$ and $y_m(kT_s, \overline{\theta})$, respectively, $\overline{\theta}$ is the vector of process model parameters, $N_S$ is the number of collected samples, $T_S$ is the sampling period, and $N_S T_S$ is the time length of the dynamic (transient) response.

The Mean Absolute Error (MAE), the expression of which is (34), has also been calculated in the following examples for illustrative purposes.

$$E(\overline{\theta}) = \frac{1}{N_s} \sum_{k=1}^{N_s} \left| e(kT_s, \overline{\theta}) \right| = \frac{1}{N_s} \sum_{k=1}^{N_s} \left| y(kT_s) - y_m(kT_s, \overline{\theta}) \right| \tag{34}$$

In the same context, it may be interesting to evaluate the goodness of fit of the identified model at different intervals of the reaction curve. For this reason, the index $S_{x_i-x_j}(\overline{\theta})$ is introduced in this paper, where $\overline{\theta}$ is the vector of identified model parameters. This performance index represents the time-domain performance index $S(\overline{\theta})$ restricted to the interval $[x_i-x_j]$ on the reaction curve, where $x_i$ and $x_j$ are two specific points on the reaction curve as depicted in Figure 3. Note that if the step response of the identified model is divided into p intervals $[x_i-x_j]$, the model performance index $S(\overline{\theta})$ and the p performance indices $S_k(\overline{\theta})$ ($k = 1, \ldots, p$) of each of the intervals satisfy the following expression:

$$S(\overline{\theta}) = \frac{1}{N_s} \sum_{k=1}^{p} S_k(\overline{\theta}) \cdot N_{sk} \tag{35}$$

where $p \in \mathbb{Z}^+$ is the number of intervals into which the model step response is divided, and $N_{Sk}$ is the number of samples of the model step response in the corresponding k interval.

The simulation results obtained in this section have been performed using the FOTF MATLAB toolbox, which is a set of built-in functions that extends the control toolbox to deal with fractional-order systems [43]. For a deeper knowledge of the FOTF toolbox, the reader is referred to the reference text in [44].

*4.1. Example 1*

In this example, the fractional-order process model (30) is selected. This model is a lag-dominated fractional second-order process plus dead-time (FSOPDT), providing some modelling deviation from the model structure selected in the proposed identification method, which is the FFOPDT dynamics.

This same model but for different $\alpha$-values in the range $0.60 \le \alpha \le 1.00$ has been used in [39], on the one hand, as a batch of processes to validate the proposed identification procedure for different sets of symmetrical points and, on the other hand, to obtain insight into the influence of the location of symmetrical points on the accuracy of the identified model.

The objective in this case is to validate the identification procedure for asymmetrical points on the process reaction curve and to obtain insight into the influence of the location of such asymmetrical points on the accuracy of the fractional-order identified model. A comparison between the proposed method and several well-known identification methods for integer-order models is also provided in this section.

The process reaction curve for this model and the step-input signal are shown in Figure 7.

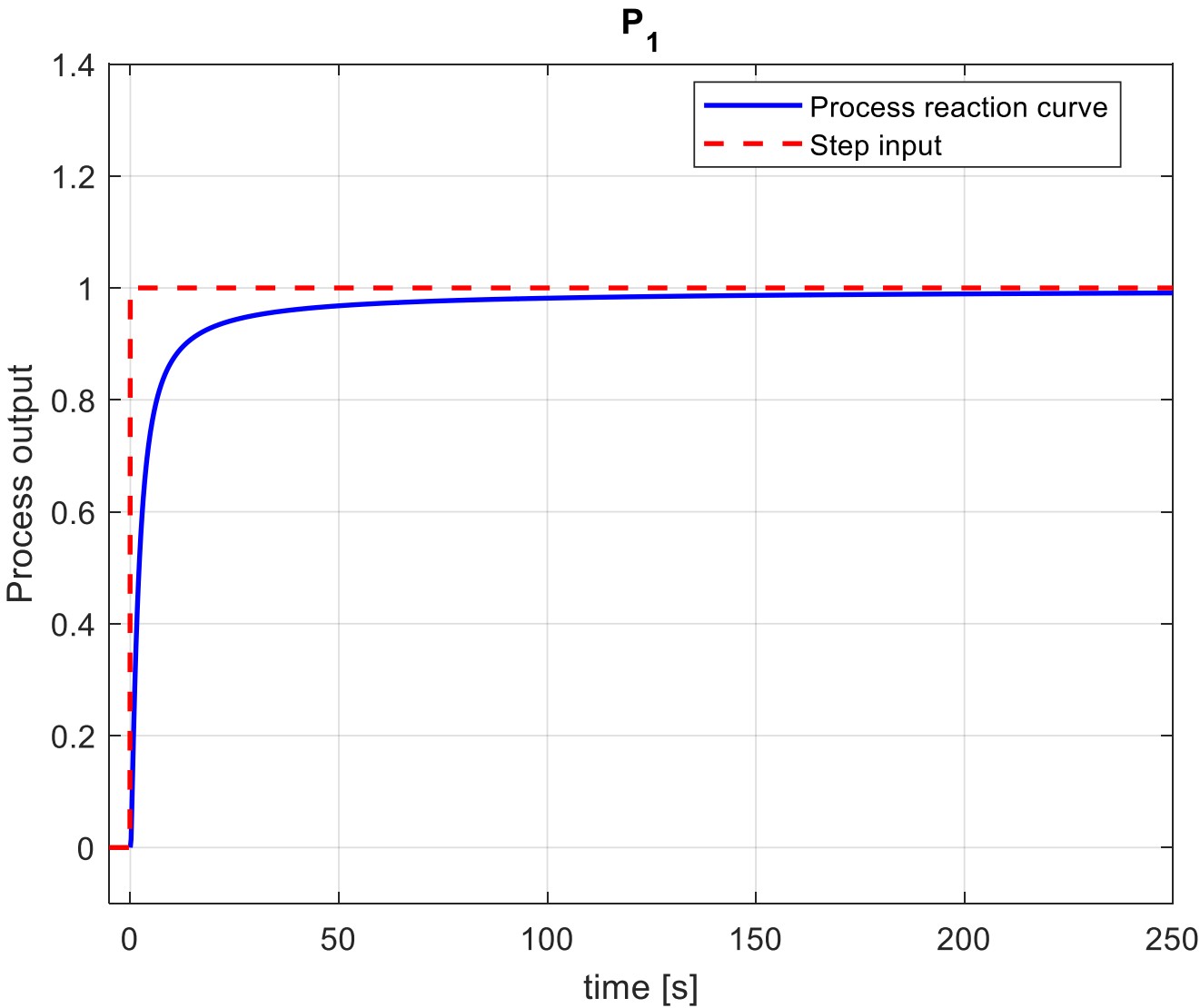

**Figure 7.** Process reaction curve for process P$_1$ and step-input signal.

The process information summarized in Table 10 for the proposed identification method is collected from data in Figure 7.

With the information collected from Table 10, the following FFOPDT model parameters $\bar{\theta}_{1,i} = \{K_{1,i}, T_{1,i}, L_{1,i}, \alpha_{1,i}\}$, for $i = 1, \ldots, 6$, have been obtained in Table 11 for the different sets of points proposed in Tables 2 and 6.

**Table 10.** Process information collected from the reaction curve for fractional-order model identification of process P$_1$.

| | Symmetrical Methods | | | Asymmetrical Methods | |
|---|---|---|---|---|---|
| **Method #1:** (5-50-95%) | **Method #2:** (10-50-90%) | **Method #3:** (25-50-75%) | **Method #4:** (10-55-90%) | **Method #5:** (20-60-95%) | **Method #6:** (20-75-95%) |
| | | $\Delta u = 1.00$ | | | |
| | | $\Delta y = 1.00$ | | | |
| $t_5 = 0.3020$ s | $t_{10} = 0.4540$ s | $t_{25} = 0.9300$ s | $t_{10} = 0.4540$ s | $t_{20} = 0.7620$ s | $t_{20} = 0.7620$ s |

**Table 10.** *Cont.*

| | Symmetrical Methods | | | Asymmetrical Methods | |
|---|---|---|---|---|---|
| **Method #1:**<br>**(5-50-95%)** | **Method #2:**<br>**(10-50-90%)** | **Method #3:**<br>**(25-50-75%)** | **Method #4:**<br>**(10-55-90%)** | **Method #5:**<br>**(20-60-95%)** | **Method #6:**<br>**(20-75-95%)** |
| $t_{50} = 2.0910$ s | $t_{50} = 2.0910$ s | $t_{50} = 2.0910$ s | $t_{55} = 2.4400$ s | $t_{60} = 2.8590$ s | $t_{75} = 4.9730$ s |
| $t_{95} = 29.1410$ s | $t_{90} = 13.2640$ s | $t_{75} = 4.9730$ s | $t_{90} = 13.2640$ s | $t_{95} = 29.1410$ s | $t_{95} = 29.1410$ s |

**Table 11.** Fractional-order model settings for the symmetrical and asymmetrical sets of points (5-50-95%), (10-50-90%), (25-50-75%), (10-55-90%), (20-60-95%), and (20-75-95%), respectively, applied to Example 1.

| | Symmetrical Methods | | | Asymmetrical Methods | |
|---|---|---|---|---|---|
| **Method #1:**<br>**(5-50-95%)** | **Method #2:**<br>**(10-50-90%)** | **Method #3:**<br>**(25-50-75%)** | **Method #4:**<br>**(10-55-90%)** | **Method #5:**<br>**(20-60-95%)** | **Method #6:**<br>**(20-75-95%)** |
| $K_{1,1} = 1.0000$ | $K_{1,2} = 1.0000$ | $K_{1,3} = 1.0000$ | $K_{1,4} = 1.0000$ | $K_{1,5} = 1.0000$ | $K_{1,6} = 1.0000$ |
| $T_{1,1} = 2.3137$ s | $T_{1,2} = 2.2492$ s | $T_{1,3} = 2.1890$ s | $T_{1,4} = 2.1963$ s | $T_{1,5} = 2.0223$ s | $T_{1,6} = 1.8807$ s |
| $L_{1,1} = 0.2745$ s | $L_{1,2} = 0.2782$ s | $L_{1,3} = 0.3901$ s | $L_{1,4} = 0.2791$ s | $L_{1,5} = 0.3139$ s | $L_{1,6} = 0.2921$ s |
| $\alpha_{1,1} = 0.7791$ | $\alpha_{1,2} = 0.7888$ | $\alpha_{1,3} = 0.8088$ | $\alpha_{1,4} = 0.7836$ | $\alpha_{1,5} = 0.7580$ | $\alpha_{1,6} = 0.7463$ |

The FFOPDT model step responses for (5-50-95%), (10-50-90%), (25-50-75%), (10-55-90%), (20-60-95%), and (20-75-95%), respectively, are compared with the process reaction curve and illustrated in Figures 8–10. In each figure the corresponding representative points, symmetrical (x-50-(100−x)%) and asymmetrical (x$_1$-x$_2$-x$_3$%), respectively, are also displayed.

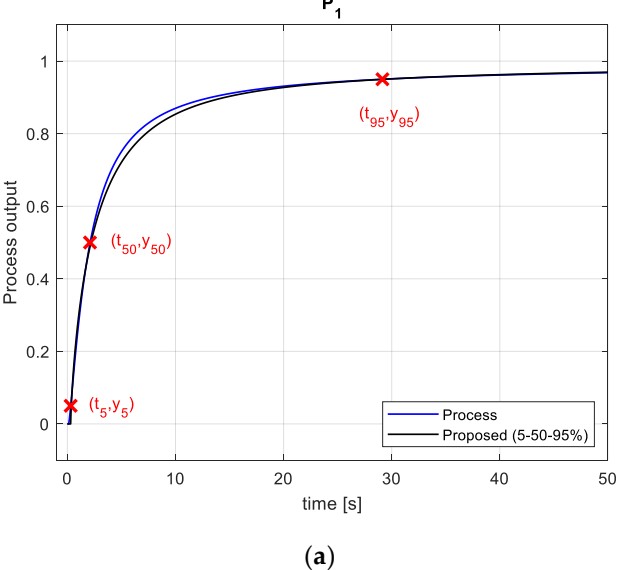

(**a**)

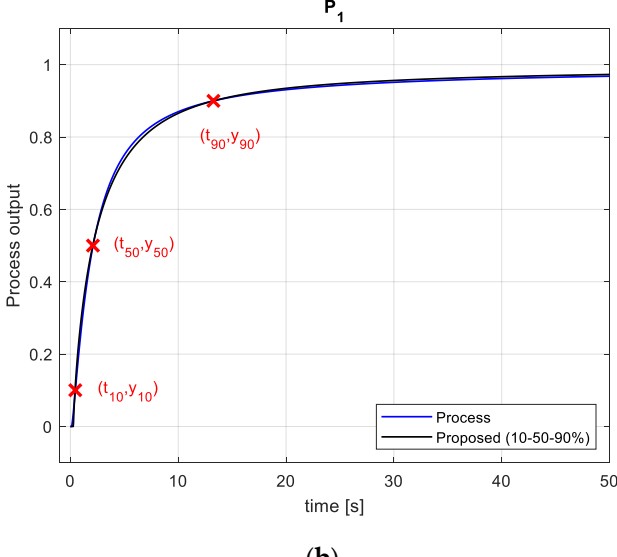

(**b**)

**Figure 8.** FFOPDT model step response using the proposed identification method for process P$_1$ and process reaction curve: (**a**) Symmetrical set of points (5-50-95%); (**b**) Symmetrical set of points (10-50-90%).

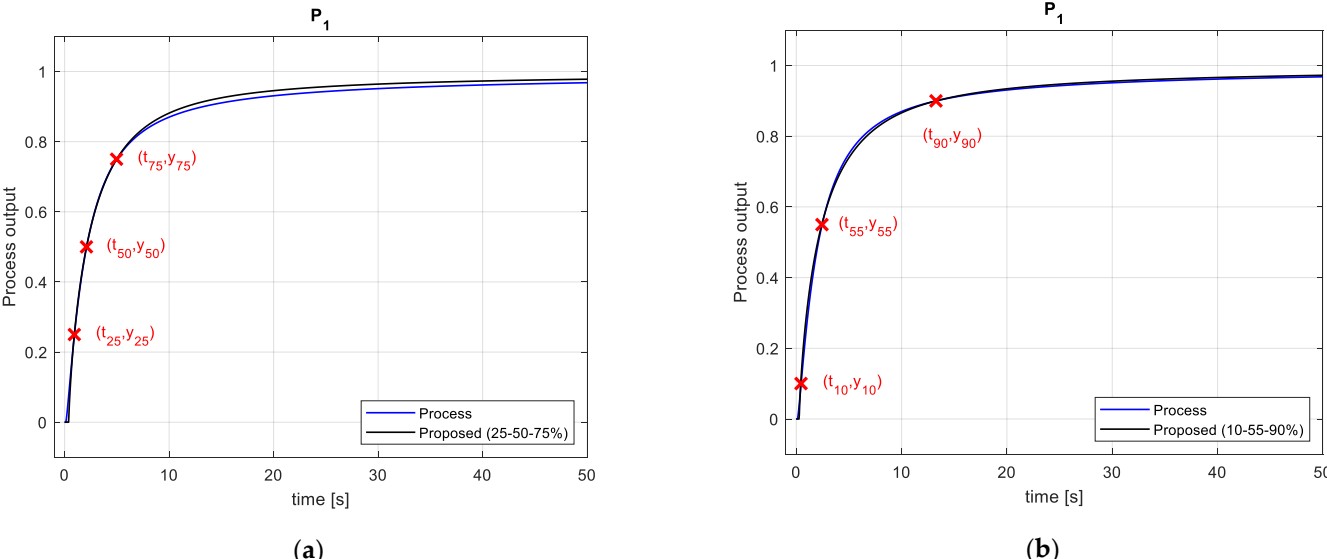

**Figure 9.** FFOPDT model step response using the proposed identification method for process $P_1$ and process reaction curve: (**a**) Symmetrical set of points (25-50-75%); (**b**) Asymmetrical set of points (10-55-90%).

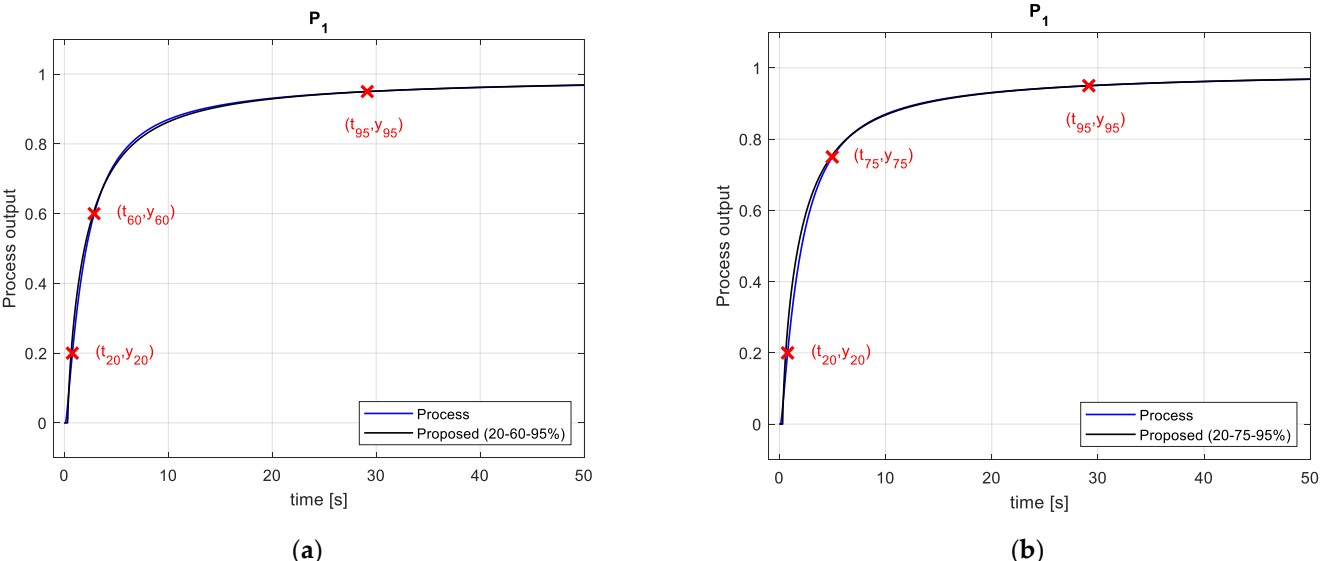

**Figure 10.** FFOPDT model step response using the proposed identification method for process $P_1$ and process reaction curve: (**a**) Asymmetrical set of points (20-60-95%); (**b**) Asymmetrical set of point (20-75-95%).

Figures 8–10 show that the step responses of the fractional-order models identified with the proposed method, for both the symmetrical and asymmetrical case, give good fit with the process reaction curve, which confirms the validity of this identification method also for the asymmetrical case. Note that the identification method for the symmetrical case has been studied in detail and its validity for the identification of FFOPDT models has also been confirmed in [39].

Table 12 shows the process parameters identified for FOPDT and DPPDT models obtained using the methods proposed by Alfaro in [11], and by Viteckóvá et al. in [14], and the ones for SOPDT obtained using methods proposed by Stark in [16] and by Jahanmiri and Fallahi in [15], using two or three points from the process reaction curve.

**Table 12.** FOPDT, DPPDT, and SOPDT model settings obtained for the considered integer-order identification methods.

| | FOPDT | | DPPDT | | SOPDT | |
|---|---|---|---|---|---|---|
| | **Alfaro [11] (25–75%)** | **Vitecková [14] (33–70%)** | **Alfaro [11] (25–75%)** | **Vitecková [14] (33–70%)** | **Stark [16] (15-45-75%)** | **Jahanmiri– Fallahi [15] (2-70-90%)** |
| | K = 1.00 | K = 1.00 | K = 1.00 | K = 1.00 | K = 1.0000 | $K_{1,6}$ = 1.0000 |
| | T = 3.68 s | T = 3.51 s | T = 2.24 s | T = 2.33 s | $T_1$ = 3.52 s | $T_1$ = 5.61 s |
| | L = 0.00 s | L = 0.00 s | L = 0.00 s | L = 0.00 s | $T_2$ = 0.34 s | $T_2$ = 0.0072 s |
| | - | - | - | - | L = 0.00 s | L = 0.20 s |

The corresponding integer-order model step responses for the considered classical identification methods are compared with the process reaction curve and illustrated in Figure 11 for FOPDT and DPPDT models and in Figure 12 for SOPDT models.

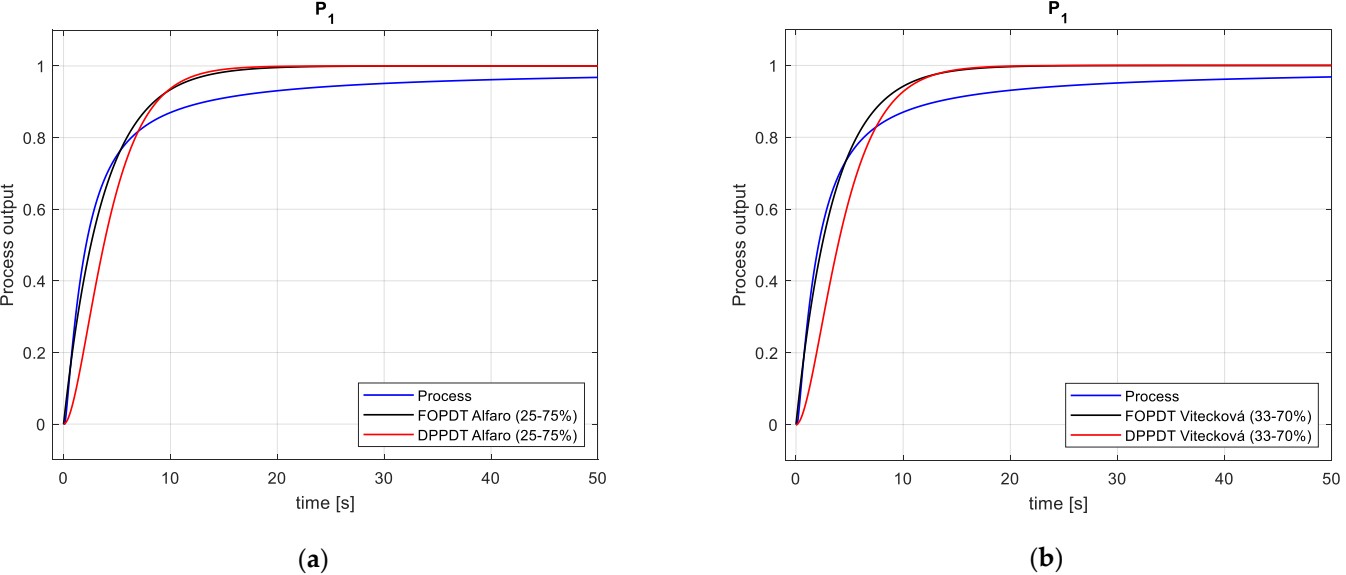

(**a**)                                                 (**b**)

**Figure 11.** FOPDT and DPPDT model step responses using integer-order model identification methods for process $P_1$ and process reaction curve: (**a**) Alfaro [11] (25–75%); (**b**) Vitecková et al. [14] (33–70%).

Figures 11 and 12 illustrate that FOPDT, DPPDT, and SOPDT models, respectively, approximate process $P_1$ with insufficient accuracy compared to FFOPDT models obtained with the proposed method. It has been illustrated from Figures 8–12 that the proposed identification method outperforms significantly the considered methods for integer-order models.

Table 13 shows the values of the time-domain performance indexes $S(\overline{\theta}_{1,i})$ and $E(\overline{\theta}_{1,i})$ for process $P_1$, and the ones corresponding to the intervals [0–50%], $S_{0-50}(\overline{\theta}_{1,i})$, and [50–100%], $S_{50-100}(\overline{\theta}_{1,i})$, of the total process output change, respectively, for the different models considered in this example. In this table, i = 1, . . . , 6 represents the different sets of points considered in the proposed identification method, and i = 7, . . . , 12 represents the different identification methods for integer-order models.

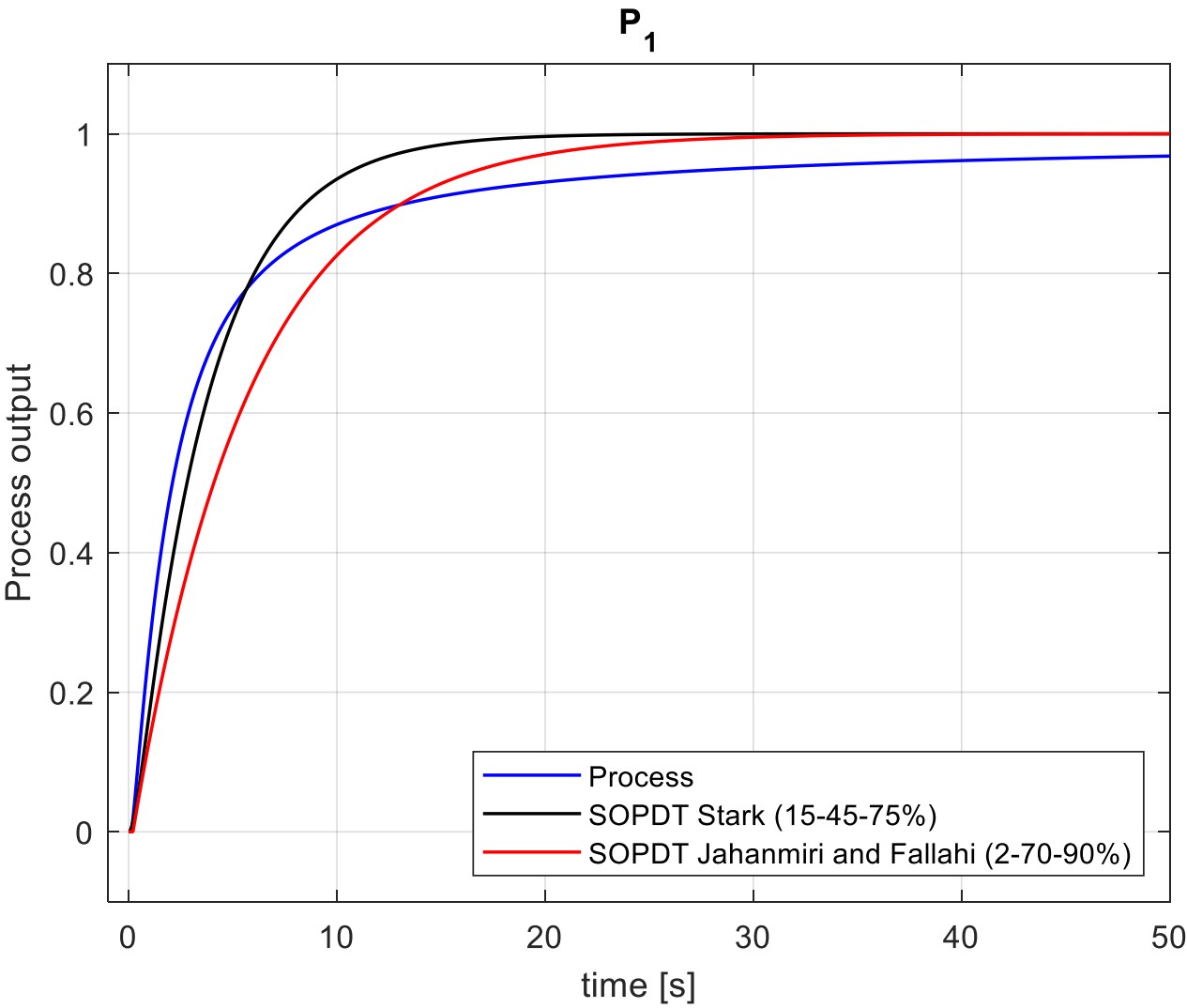

**Figure 12.** SOPDT model step responses using methods proposed by Stark [16] (15-45-75%) and by Jahanmiri and Fallahi [15] (2-70-90%) and process reaction curve.

**Table 13.** Comparison between the performance indexes obtained for the proposed method with different sets of points, symmetrical and asymmetrical, and the ones for several well-known methods for integer-order models. The number of samples $N_S$ for the whole range and for each interval are also displayed.

| i | Method | Set of Points | $\overline{\mathbf{S}}_{\mathbf{0-50}}(\theta_{1,i})$ | $\overline{\mathbf{S}}_{\mathbf{50-100}}(\theta_{1,i})$ | $\overline{\mathbf{S}}_{\mathbf{0-100}}(\theta_{1,i}) = \overline{\mathbf{S}}(\theta_{1,i})$ | $\overline{\mathbf{E}}(\theta_{1,i})$ |
|---|---|---|---|---|---|---|
| 1 | FFOPDT Proposed #1 | (5-50-95%) | $3.07 \times 10^{-4}$ | $2.24 \times 10^{-5}$ | $2.48 \times 10^{-5}$ | $2.4 \times 10^{-3}$ |
| 2 | FFOPDT Proposed #2 | (10-50-90%) | $4.63 \times 10^{-4}$ | $1.45 \times 10^{-5}$ | $1.83 \times 10^{-5}$ | $3.6 \times 10^{-3}$ |
| 3 | FFOPDT Proposed #3 | (25-50-75%) | $4.75 \times 10^{-4}$ | $5.36 \times 10^{-5}$ | $5.72 \times 10^{-5}$ | $6.7 \times 10^{-3}$ |
| 4 | FFOPDT Proposed #4 | (10-55-90%) | $6.86 \times 10^{-4}$ | $1.09 \times 10^{-5}$ | $1.65 \times 10^{-5}$ | $3.2 \times 10^{-3}$ |
| 5 | FFOPDT Proposed #5 | (20-60-95%) | $1.30 \times 10^{-3}$ | $3.46 \times 10^{-6}$ | $1.44 \times 10^{-5}$ | $1.2 \times 10^{-3}$ |
| 6 | FFOPDT Proposed #6 | (20-75-95%) | $3.20 \times 10^{-3}$ | $6.86 \times 10^{-6}$ | $3.34 \times 10^{-5}$ | $9.7 \times 10^{-4}$ |
| 7 | FOPDT Alfaro | (25-75%) | - | - | $7.32 \times 10^{-4}$ | $2.19 \times 10^{-2}$ |
| 8 | FOPDT Vitecková | (33-70%) | - | - | $7.54 \times 10^{-4}$ | $2.20 \times 10^{-2}$ |
| 9 | DPPDT Alfaro | (25-75%) | - | - | $1.50 \times 10^{-3}$ | $2.51 \times 10^{-2}$ |

**Table 13.** *Cont.*

| i | Method | Set of Points | $\overline{S}_{0-50}(\overline{\theta}_{1,i})$ | $\overline{S}_{50-100}(\overline{\theta}_{1,i})$ | $\overline{S}_{0-100}(\overline{\theta}_{1,i}) = \overline{S}(\overline{\theta}_{1,i})$ | $\overline{E}(\overline{\theta}_{1,i})$ |
|---|---|---|---|---|---|---|
| 10 | DPPDT Vitecková | (33-70%) | - | - | $1.60 \times 10^{-3}$ | $2.53 \times 10^{-2}$ |
| 11 | SOPDT Stark | (15-45-75%) | - | - | $8.26 \times 10^{-4}$ | $2.26 \times 10^{-2}$ |
| 12 | SOPDT Jahanmiri–Fallahi | (2-70-90%) | - | - | $1.40 \times 10^{-3}$ | $2.35 \times 10^{-2}$ |
| | Number of samples | | $N_{S1} = 210$ | $N_{S2} = 24{,}791$ | $N_S = N_{S1} + N_{S2} = 25{,}001$ | $N_S = 25{,}001$ |

Time-domain performance indexes in both intervals, $S_{0-50}(\overline{\theta}_{1,i})$ and $S_{50-100}(\overline{\theta}_{1,i})$, give information about the effect of the location of the different set of points on the accuracy of the identified models. Since the step response of the identified models can be divided into the two aforementioned intervals, the accuracy of the identified models can be quantified and the one for the first and second half of the total interval can be also determined from data in Table 13. The number of samples for the model step responses in the whole range, $N_S$, and in both intervals, $N_{S1}$ and $N_{S2}$, respectively, are also provided in Table 13.

Note that from expression (35) the following relation that must be fulfilled can be particularized for this case:

$$S(\overline{\theta}_{1,i}) = S_{0-100}(\overline{\theta}_{1,i}) = \frac{1}{N_s}\left[S_{0-50}(\overline{\theta}_{1,i}) \cdot N_{s1} + S_{50-100}(\overline{\theta}_{1,i}) \cdot N_{s2}\right] \tag{36}$$

The results of Table 13 for the proposed identification method in terms of $S_{0-50}(\overline{\theta}_{1,i})$, $S_{50-100}(\overline{\theta}_{1,i})$, and $S(\overline{\theta}_{1,i})$, for i = 1, ... , 6, are also shown graphically in Figure 13.

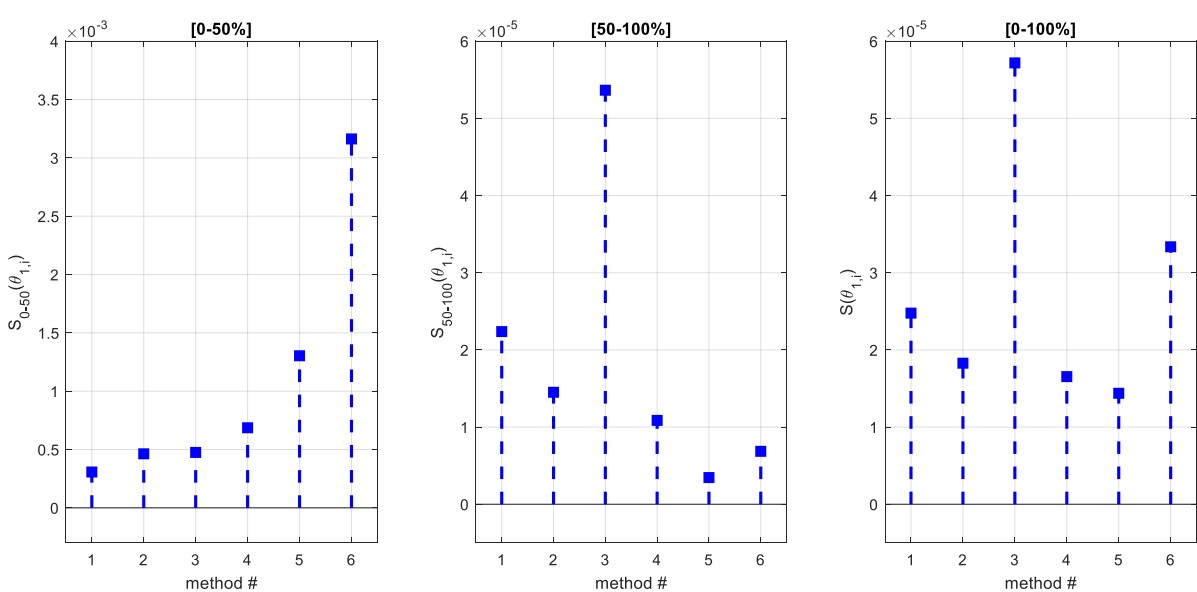

**Figure 13.** Graphical representation of the data given in Table 13 for process $P_1$. From left to right the performance indexes $S_{0-50}(\overline{\theta}_{1,i})$, $S_{50-100}(\overline{\theta}_{1,i})$, and $S(\overline{\theta}_{1,i})$, i = 1, ... , 6, are shown, respectively, for each of the considered methods.

Next, the results obtained in Table 13 and illustrated in Figure 13 for the FFOPDT models identified for process $P_1$ will be analyzed in order to gain insights into the location of the points for the asymmetrical case and their effect on the accuracy of the identified model. The following observations can be drawn from the above results:

1. Methods with low values of $x_1$ for the asymmetrical case, or x in the symmetrical case, allow obtaining low values of $S_{0-50}$, as can be seen in Figure 13 for methods #1, #2, and #4, compared to #5 and #6, which give worse results in that interval.

2. Methods with high values of $x_3$ for the asymmetrical case allow better fitting of the model to the reaction curve in the final part of the response, as verified by the low values of $S_{50-100}$ for methods #4, #5, and #6. For the symmetrical case, a high value of $(100 - x)$ implies a low value of $x$, thus better fitting the model to the reaction curve in the initial and final parts of the response, as illustrated by the low values of $S_{0-50}$ and $S_{50-100}$, respectively, for methods #1 and #2.

3. Comparing methods #2 and #4, it can be seen that the effect of increasing the value of $x_2$, while keeping $x_1$ and $x_3$ constant, is to reduce the value of S as a result of a reduction in the value of $S_{50-100}$, even though the value of $S_{0-50}$ is slightly increased.

4. Comparing methods #5 and #6, the effect of increasing the value of $x_2$ can also be observed, while keeping $x_1$ and $x_3$ constant. Note that the value of $x_1$ in this case is higher than for methods #2 and #4. The value of S for method #5 is reduced as a result of a very good fit in the interval [50–100%], as shown in Figure 13. The value of S corresponding to method #6 increases substantially due to a poorer fit in the interval [0–50%] despite the good result for $S_{50-100}$.

As discussed previously, the step response of the identified model can be divided into different intervals to evaluate the effect that the selection of different sets of points ($x_1$-$x_2$-$x_3$%) has on the model's accuracy in each interval.

In this regard, Table 14 allows the comparison of method #2 with #4 for the intervals [0–10%], [10–55%], [55–90%], and [90–100%], respectively. The comparison of methods #5 and #6 for the intervals [0–20%], [20–50%], [50–75%], and [75–100%], respectively, is shown in Table 15.

**Table 14.** Comparison between methods #2 and #4 in terms of time-domain model performance indices for different intervals of the process reaction curve for both identified models. The time-domain performance index for the whole response is also included in this table. $N_S$ represents the number of points for each interval.

| Interval # | Interval | Method #2: (10-50-90%) | Method #4: (10-55-90%) | $N_S$ |
|---|---|---|---|---|
| 1 | [0–10%] | $S_{0-10}(\bar{\theta}_{1,2}) = 2.61 \times 10^{-4}$ | $S_{0-10}(\bar{\theta}_{1,4}) = 2.64 \times 10^{-4}$ | 46 |
| 2 | [10–55%] | $S_{10-55}(\bar{\theta}_{1,2}) = 4.31 \times 10^{-4}$ | $S_{10-55}(\bar{\theta}_{1,4}) = 6.76 \times 10^{-4}$ | 198 |
| 3 | [55–90%] | $S_{55-90}(\bar{\theta}_{1,2}) = 8.11 \times 10^{-5}$ | $S_{55-90}(\bar{\theta}_{1,4}) = 4.90 \times 10^{-5}$ | 1117 |
| 4 | [90–100%] | $S_{90-110}(\bar{\theta}_{1,2}) = 1.14 \times 10^{-5}$ | $S_{90-100}(\bar{\theta}_{1,4}) = 9.06 \times 10^{-6}$ | 23,674 |
| - | [0–100%] | $S(\bar{\theta}_{1,2}) = 1.83 \times 10^{-5}$ | $S(\bar{\theta}_{1,4}) = 1.65 \times 10^{-5}$ | 25,001 |

**Table 15.** Comparison between methods #5 and #6 in terms of time-domain model performance indices for different intervals of the process reaction curve for both identified models. The time-domain performance index for the whole response is also included in this table. $N_S$ represents the number of points for each interval.

| Interval # | Interval | Method #5: (20-60-95%) | Method #6: (20-75-95%) | $N_S$ |
|---|---|---|---|---|
| 1 | [0–20%] | $S_{0-20}(\bar{\theta}_{1,5}) = 7.65 \times 10^{-4}$ | $S_{0-20} = 1.60 \times 10^{-3}$ | 77 |
| 2 | [20–50%] | $S_{20-50}(\bar{\theta}_{1,5}) = 1.60 \times 10^{-3}$ | $S_{20-50}(\bar{\theta}_{1,6}) = 4.10 \times 10^{-3}$ | 133 |
| 3 | [50–75%] | $S_{50-75}(\bar{\theta}_{1,5}) = 9.31 \times 10^{-5}$ | $S_{50-75}(\bar{\theta}_{1,6}) = 5.69 \times 10^{-4}$ | 288 |
| 4 | [75–100%] | $S_{75-100}(\bar{\theta}_{1,5}) = 2.40 \times 10^{-6}$ | $S_{75-100}(\bar{\theta}_{1,6}) = 2.45 \times 10^{-7}$ | 24,503 |
| - | [0–100%] | $S(\bar{\theta}_{1,5}) = 1.44 \times 10^{-5}$ | $S(\bar{\theta}_{1,6}) = 3.34 \times 10^{-5}$ | 25,001 |

### 4.2. Example 2

The model (31) proposed in [31] is considered in this example. This process exhibits a higher-order lag-dominated fractional-order dynamics. The proposed identification method for symmetrical and asymmetrical sets of points are compared with other well-recognized methods for FFOPDT models, which are also based on the process reaction curve, and the optimal FOPDT.

The step-input signal and the process reaction curve for this model are shown in Figure 14.

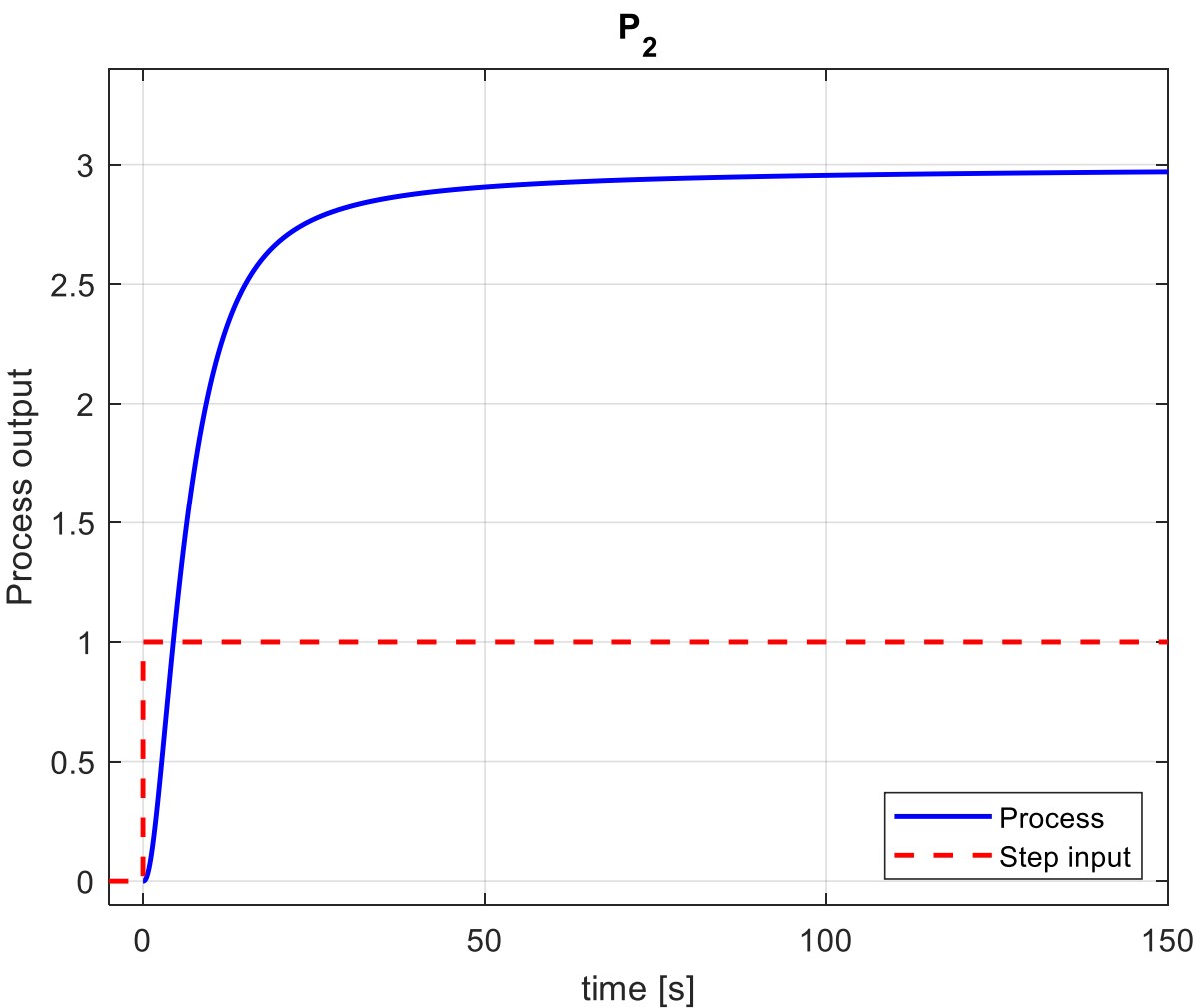

**Figure 14.** Process reaction curve for process $P_2$ and step-input signal.

The process information summarized in Table 16 for the different identification methods is collected from data in Figure 14.

**Table 16.** Process information collected from the reaction curve for fractional-order model identification of process $P_2$.

| Symmetrical Methods | | | Asymmetrical Methods | | |
|---|---|---|---|---|---|
| Method #1: (5-50-95%) | Method #2: (10-50-90%) | Method #3: (25-50-75%) | Method #4: (10-55-90%) | Method #5: (20-60-95%) | Method #6: (20-75-95%) |
| $\Delta u = 1.00$ | | | | | |
| $\Delta y = 3.00$ | | | | | |
| $t_5 = 1.4140$ s | $t_{10} = 2.0290$ s | $t_{25} = 3.5600$ s | $t_{10} = 2.0290$ s | $t_{20} = 3.0630$ s | $t_{20} = 3.0630$ s |
| $t_{50} = 6.3850$ s | $t_{50} = 6.3850$ s | $t_{50} = 6.3850$ s | $t_{55} = 7.1044$ s | $t_{60} = 7.9180$ s | $t_{75} = 11.4060$ s |
| $t_{95} = 33.9290$ s | $t_{90} = 20.8030$ s | $t_{75} = 11.4060$ s | $t_{90} = 20.8030$ s | $t_{95} = 33.9290$ s | $t_{95} = 33.9290$ s |

With the information collected from Table 16, the following FFOPDT model parameters $\bar{\theta}_{2,i} = \{K_{2,i}, T_{2,i}, L_{2,i}, \alpha_{2,i}\}$, for i = 1, ... , 6, have been obtained in Table 17 for the different sets of points, i.e., (5-50-95%), (10-50-90%), (25-50-75%), (10-55-90%), (20-60-95%), and (20-75-95%), respectively.

**Table 17.** Fractional-order model settings for the symmetrical and asymmetrical sets of points (5-50-95%), (10-50-90%), (25-50-75%), (10-55-90%), (20-60-95%), and (20-75-95%), respectively, applied to Example 2.

| Symmetrical Methods | | | Asymmetrical Methods | | |
|---|---|---|---|---|---|
| **Method #1: (5-50-95%)** | **Method #2: (10-50-90%)** | **Method #3: (25-50-75%)** | **Method #4: (10-55-90%)** | **Method #5: (20-60-95%)** | **Method #6: (20-75-95%)** |
| $K_{2,1} = 3.0000$ | $K_{2,2} = 3.0000$ | $K_{2,3} = 3.0000$ | $K_{2,4} = 3.0000$ | $K_{2,5} = 3.0000$ | $K_{2,6} = 3.0000$ |
| $T_{2,1} = 6.4066$ s | $T_{2,2} = 6.6381$ s | $T_{2,3} = 6.6817$ s | $T_{2,4} = 6.3850$ s | $T_{2,5} = 5.1733$ s | $T_{2,6} = 4.6914$ s |
| $L_{2,1} = 1.2638$ s | $L_{2,2} = 1.3955$ s | $L_{2,3} = 1.6203$ s | $L_{2,4} = 1.4329$ s | $L_{2,5} = 2.4042$ s | $L_{2,6} = 2.5096$ s |
| $\alpha_{2,1} = 0.9189$ | $\alpha_{2,2} = 0.9470$ | $\alpha_{2,3} = 0.9802$ | $\alpha_{2,4} = 0.9391$ | $\alpha_{2,5} = 0.8901$ | $\alpha_{2,6} = 0.8759$ |

Process (31) is also approximated by an FFOPDT model following the method proposed by Tavakoli-Kakhki in [31], by an FFOPDT model obtained using the optimization-based method proposed by Guevara et al. in [34], and by the optimal FOPDT model, where model parameters, $\overline{\theta}_{2,i}$ for i = 7, 8, and 9, respectively, are given in Table 18.

**Table 18.** FFOPDT model parameters obtained using methods proposed by Tavakoli-Kakhki [31] and by Guevara et al. [34], and optimal FOPDT model parameters, respectively, for process $P_2$.

| **Method #7: FFOPDT Tavakoli-Kakhki [31]** | **Method #8: FFOPDT Guevara et al. [34]** | **Method #9: FOPDT optimal** |
|---|---|---|
| $K_{2,7} = 3.00$ | $K_{2,8} = 3.0000$ | $K_{2,9} = 3.0000$ |
| $T_{2,7} = 6.30$ s | $T_{2,8} = 5.6285$ s | $T_{2,9} = 8.7412$ s |
| $L_{2,7} = 1.00$ s | $L_{2,8} = 1.8833$ s | $L_{2,9} = 0.0000$ s |
| $\alpha_{2,7} = 0.92$ | $\alpha_{2,8} = 0.9263$ | - |

Note that the method proposed in [34] is based on optimization, where the function to be minimized is $\overline{E}(\theta)$ (34). In this method, the approximation of the fractional term $s^\alpha$ is developed using the Oustaloup method; see [19]. In contrast, the parameters obtained in the optimal FOPDT model are those that minimize the function $\overline{S}(\theta)$ (33).

The step responses of the considered approximated models for (5-50-95%), (10-50-90%), (25-50-75%), (10-55-90%), (20-60-95%), and (20-75-95%), respectively, are compared with the process reaction curve and illustrated in Figures 15–17. The corresponding representative points, symmetrical (x-50-(100 − x)%) and asymmetrical ($x_1$-$x_2$-$x_3$%), respectively, are also displayed in these figures. Moreover, the process reaction curve and the step responses of the FFOPDT model proposed by Tavakoli-Kakhki, the FFOPDT model obtained using the method proposed by Guevara et al., and the optimal FOPDT model, respectively, are shown in Figure 18.

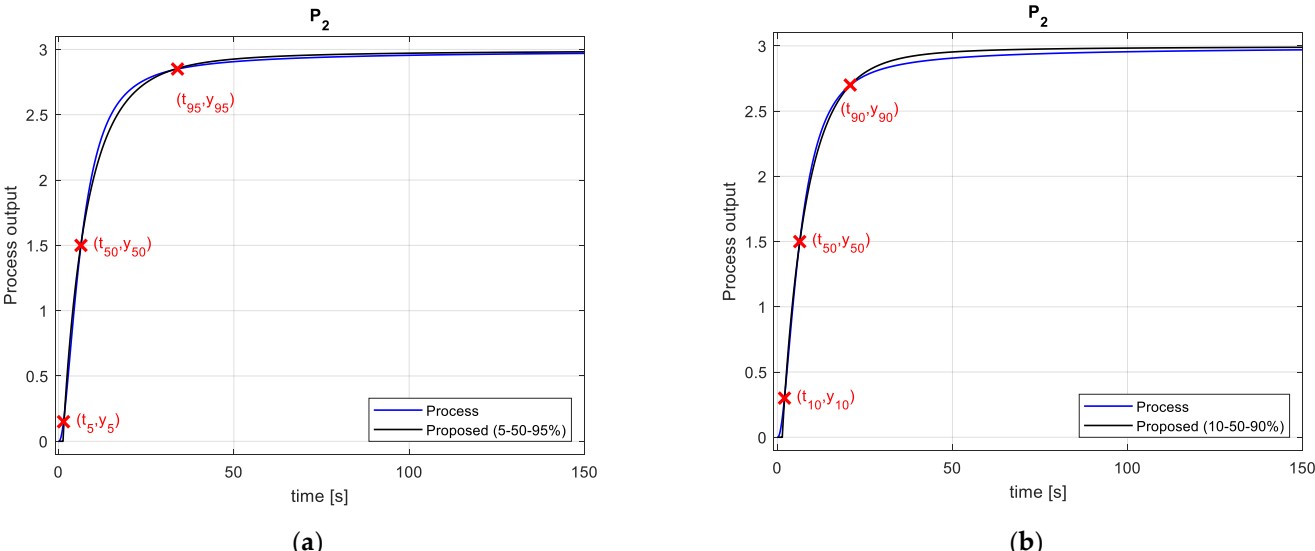

**Figure 15.** FFOPDT model step response using the proposed identification method for process $P_2$ and process reaction curve: (**a**) Symmetrical set of points (5-50-95%); (**b**) Symmetrical set of points (10-50-90%).

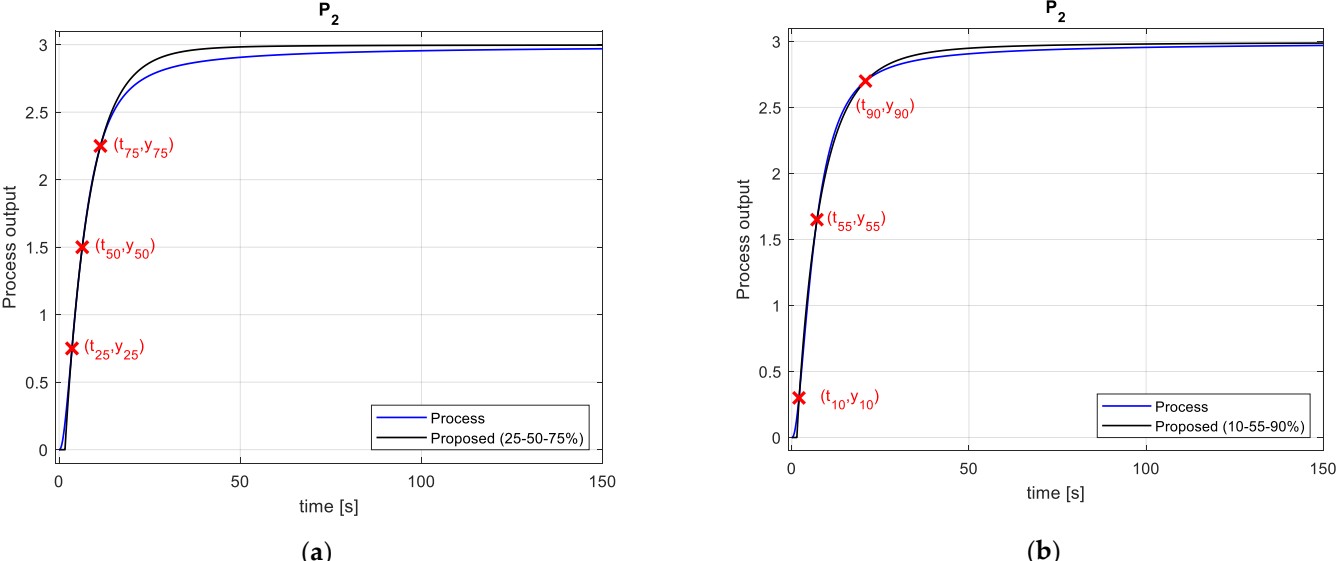

**Figure 16.** FFOPDT model step response using the proposed identification method for process $P_2$ and process reaction curve: (**a**) Symmetrical set of points (25-50-75%); (**b**) Asymmetrical set of points (10-55-90%).

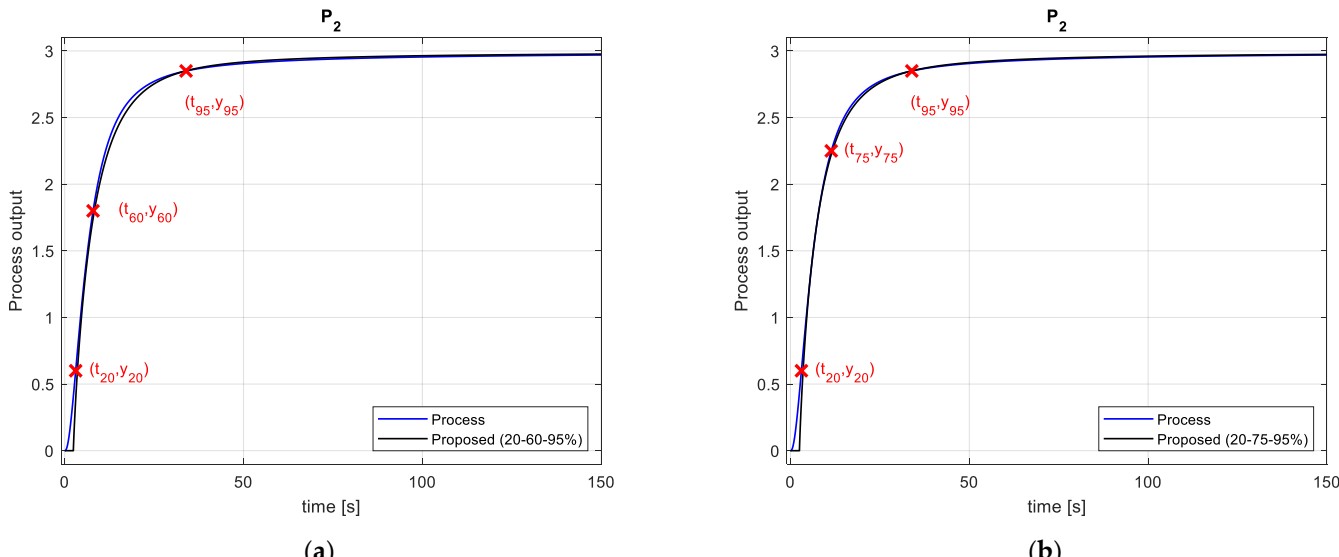

(**a**)　　　　　　　　　　　　　　(**b**)

**Figure 17.** FFOPDT model step response using the proposed identification method for process $P_2$ and process reaction curve: (**a**) Asymmetrical set of points (20-60-95%); (**b**) Asymmetrical set of points (20-75-95%).

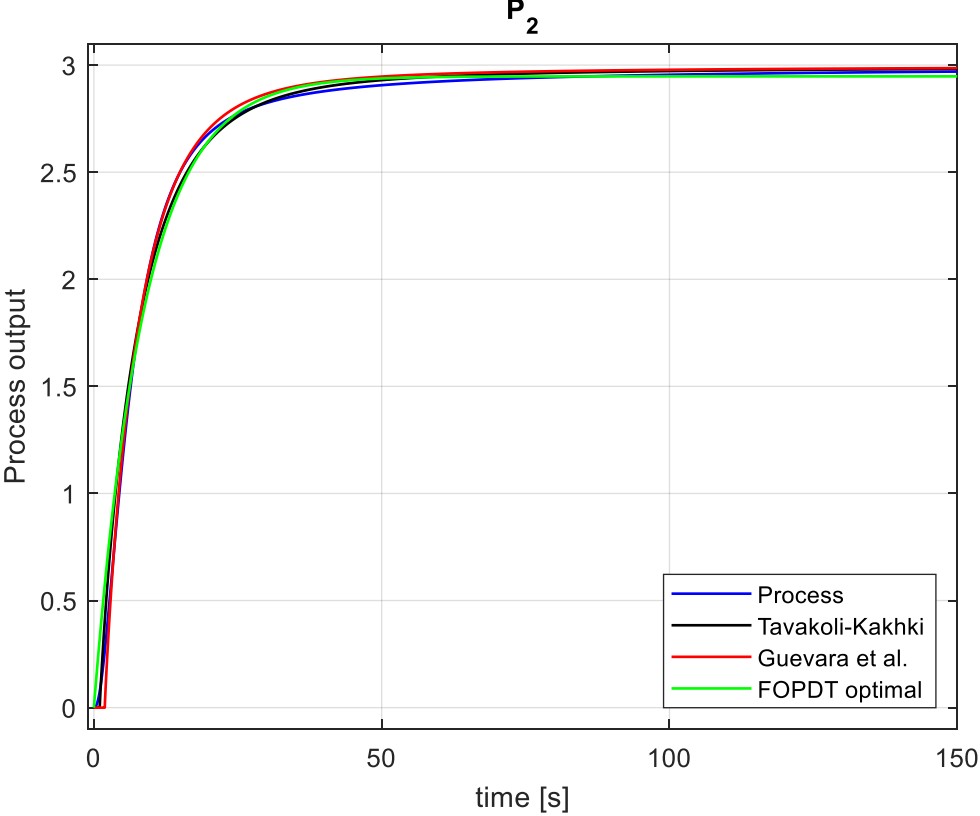

**Figure 18.** Step response of the FFOPDT model obtained using methods proposed by Tavakoli-Kakhki [31] and by Guevara [34], respectively, the step response of the optimal FOPDT model, for process $P_2$ and process reaction curve.

In this example, a high-order process model has been utilized to demonstrate the effectivity and applicability of the proposed identification procedure in the task of estimating FFOPDT-type model parameters.

Similar to the previous example, Figures 15–17 illustrate that the models obtained with the proposed method, for both the symmetrical and asymmetrical case, provide a good fit with the process reaction curve also for a higher-order lag-dominated fractional-order process in comparison with the results obtained using the well-recognized method proposed by Tavakoli-Kakhki, and even in comparison with optimization-based methods for FFOPDT and FOPDT models. These results are shown in Figure 18.

Figure 15a,b and Figure 16a show that the proposed method for the symmetrical case gives good results in the interval [x-(100 − x)%]. For this reason, better results are obtained for low values of x, as discussed in [39]. Figures 16b and 17a,b show that by moving the points $x_1$, $x_2$, and $x_3$ of the reaction curve it is possible to obtain an FFOPDT model of which the step response can be better fitted to the reaction curve in certain intervals, as will be shown below.

Table 19 shows the values of the time-domain performance indexes $S(\bar{\theta}_{2,i})$ and $E(\bar{\theta}_{2,i})$ for process $P_2$, and the ones corresponding to the intervals [0–50%], $S_{0-50}(\bar{\theta}_{2,i})$, and [50–100%], $S_{50-100}(\bar{\theta}_{2,i})$, of the total process output change, respectively, for the different models considered in this example. In this table, $i = 1, \ldots, 6$ represents the different sets of points considered in the proposed identification method, and $i = 7, 8$, and 9 represents methods #7 and #8 for FFOPDT models, and #9 for FOPDT, respectively.

**Table 19.** Comparison between the performance indexes obtained for the proposed method with different sets of points, symmetrical and asymmetrical, and the ones for several methods for FFOPDT and FOPDT models. The number of samples $N_S$ for the whole range and for each interval are also displayed.

| i | Method | Set of Points | $S_{0-50}(\bar{\theta}_{2,i})$ | $S_{50-100}(\bar{\theta}_{2,i})$ | $S_{0-100}(\bar{\theta}_{2,i}) = S(\bar{\theta}_{2,i})$ | $E(\bar{\theta}_{2,i})$ |
|---|---|---|---|---|---|---|
| 1 | FFOPDT Proposed #1 | (5-50-95%) | $5.80 \times 10^{-3}$ | $8.78 \times 10^{-4}$ | $1.10 \times 10^{-3}$ | $2.40 \times 10^{-2}$ |
| 2 | FFOPDT Proposed #2 | (10-50-90%) | $2.80 \times 10^{-3}$ | $1.20 \times 10^{-3}$ | $1.30 \times 10^{-3}$ | $3.34 \times 10^{-2}$ |
| 3 | FFOPDT Proposed #3 | (25-50-75%) | $4.40 \times 10^{-3}$ | $3.33 \times 10^{-3}$ | $3.30 \times 10^{-3}$ | $5.14 \times 10^{-2}$ |
| 4 | FFOPDT Proposed #4 | (10-55-90%) | $3.80 \times 10^{-3}$ | $9.58 \times 10^{-4}$ | $1.10 \times 10^{-3}$ | $3.02 \times 10^{-2}$ |
| 5 | FFOPDT Proposed #5 | (20-60-95%) | $2.78 \times 10^{-2}$ | $4.53 \times 10^{-4}$ | $1.60 \times 10^{-3}$ | $1.92 \times 10^{-2}$ |
| 6 | FFOPDT Proposed #6 | (20-75-95%) | $2.88 \times 10^{-2}$ | $1.01 \times 10^{-4}$ | $1.30 \times 10^{-3}$ | $1.24 \times 10^{-2}$ |
| 7 | FFOPDT Tavakoli-Kakhki [31] | - | $2.02 \times 10^{-2}$ | $5.25 \times 10^{-4}$ | $1.40 \times 10^{-3}$ | $2.47 \times 10^{-2}$ |
| 8 | FFOPDT Guevara et al. [34] | - | $8.30 \times 10^{-3}$ | $8.23 \times 10^{-4}$ | $1.10 \times 10^{-3}$ | $2.82 \times 10^{-2}$ |
| 9 | FOPDT optimal | - | $4.55 \times 10^{-2}$ | $9.07 \times 10^{-4}$ | $2.80 \times 10^{-3}$ | $2.95 \times 10^{-2}$ |
| | Number of samples | | $N_{S1} = 639$ | $N_{S2} = 14362$ | $N_S = N_{S1} + N_{S2} = 15{,}001$ | $N_S = 15{,}001$ |

Note that from expression (35) the following relation that must be fulfilled can be particularized for this case:

$$S(\bar{\theta}_{2,i}) = S_{0-100}(\bar{\theta}_{2,i}) = \frac{1}{N_s}[S_{0-50}(\bar{\theta}_{2,i}) \cdot N_{s1} + S_{50-100}(\bar{\theta}_{2,i}) \cdot N_{s2}] \tag{37}$$

Table 19 shows that the accuracy of models obtained with methods #1, #5, and #6 is better in terms of E than the one obtained for methods #7 and #8. The proposed method for all the sets of points except #3 provides models with lower values of E than for the optimal FOPDT model.

For the sake of simplicity in the interpretation of results, information taken from Table 19 is depicted in Figure 19.

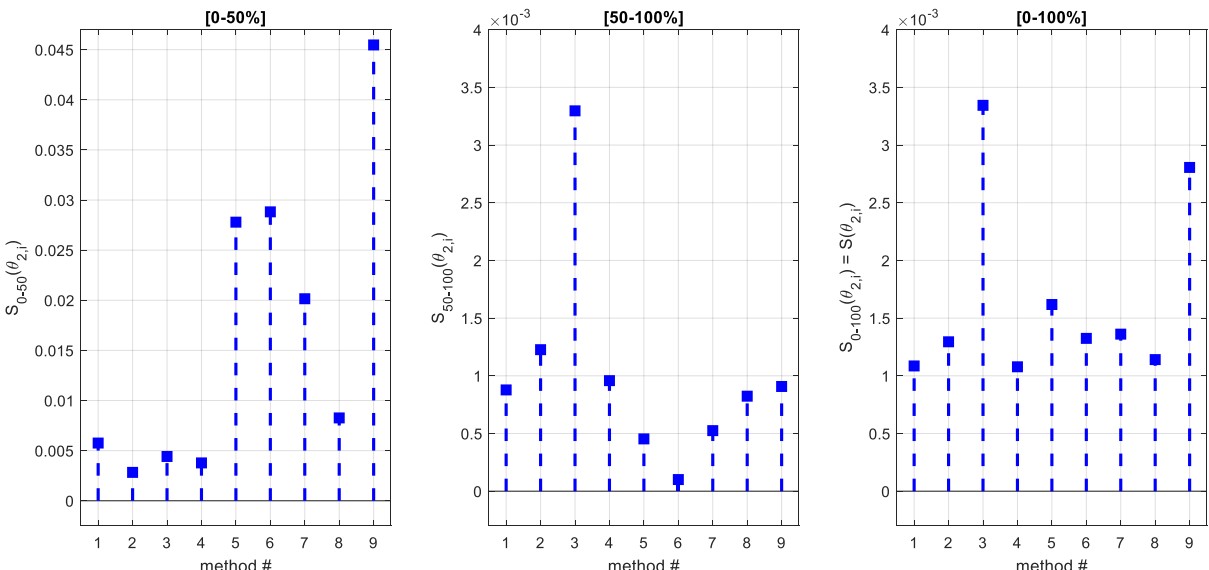

**Figure 19.** Graphical representation of data given in Table 19 for process $P_2$. From left to right the performance indices $S_{0-50}(\overline{\theta}_{2,i})$, $S_{50-100}(\overline{\theta}_{2,i})$, and $S(\overline{\theta}_{2,i})$, i = 1, ... , 9, are shown, respectively, for each of the considered methods.

Figure 19 illustrates that the accuracy of the models obtained with the proposed method for the symmetrical and asymmetrical case is similar to the one obtained using the method proposed by Guevara et al., and even better than that obtained with a well-known identification method such as the one proposed by Tavakoli-Kakhki, or than the optimal FOPDT model. Note that the method proposed by Guevara et al. is based on optimization, while the method proposed in this paper is analytical.

Specifically, the value of the time-domain model performance index of methods #1, #2, #4, and #6 are lower than the one proposed by Tavakoli-Kakhki, confirming the effectiveness of the proposed method for both the symmetrical and the asymmetrical case. Furthermore, the proposed method not only gives better results than the Tavakoli-Kakhki method in terms of S, but in the authors' opinion it is easier to apply.

The S-value of the model obtained with method #5 is slightly higher than the one by Tavakoli-Kakhki, while the one obtained with method #3 is substantially higher than the rest. This fact has already been discussed above and also in [39] for the symmetrical case and the choice of this method among the symmetrical methods allows us to determine that the accuracy of models obtained is improved for low values of x in the symmetrical case, and high values of $x_3$ for the asymmetrical case.

Results in Table 19 confirm the same observations taken from Example 1:

1.  Methods with low values of $x_1$ for the asymmetrical case, or x in the symmetrical case, present low values of $S_{0-50}$, as can be seen in Figure 19 for methods #1, #2, and #4.
2.  Methods with high values of $x_3$ for the asymmetric case present a lower value of $S_{50-100}$, as can be observed for method #4, and especially for #5 and #6. For the symmetric case, methods #1 and #2, with low values of x and high values of (100 − x), present good values of $S_{0-50}$ and $S_{50-100}$, as expected.
3.  Comparison of methods #2 and #4 yields the same conclusions as for Example 1.
4.  The observations drawn from Example 1 for the comparison of methods #5 and #6 are also extensible to Example 2.

The comparison of method #7 with the methods providing the best results in terms of S is then performed by dividing the reaction curve and the respective step responses of the obtained models into different intervals.

It can be extracted from data taken from Table 19 that the best results in terms of S are those obtained with models determined using method #1 for the symmetrical case

and #4 for the asymmetrical case. Then, the step responses of the models obtained using methods #1, #4, and #7 are divided into four intervals, i.e., [0–10%], [10–50%], [50–90%], and [90–100%], respectively. Table 20 provides a comparison of these methods for the aforementioned intervals.

**Table 20.** Comparison between methods #1, #4, and #7 in terms of time-domain model performance indexes for different intervals of the process reaction curve for both identified models. The time-domain performance index for the whole response is also included in this table. $N_S$ represents the number of points for each interval.

| Interval # | Interval | Method #1: (5-50-95%) | Method #4: (10-55-90%) | Method #7 | $N_S$ |
|---|---|---|---|---|---|
| 1 | [0–10%] | $S_{0-10}(\bar{\theta}_{2,1}) = 2.80 \times 10^{-3}$ | $S_{0-10}(\bar{\theta}_{2,4}) = 5.70 \times 10^{-3}$ | $S_{0-10}(\bar{\theta}_{2,7}) = 4.80 \times 10^{-3}$ | 203 |
| 2 | [10–50%] | $S_{10-50}(\bar{\theta}_{2,1}) = 7.20 \times 10^{-3}$ | $S_{10-50}(\bar{\theta}_{2,4}) = 2.90 \times 10^{-3}$ | $S_{10-50}(\bar{\theta}_{2,7}) = 2.73 \times 10^{-2}$ | 436 |
| 3 | [50–90%] | $S_{50-90}(\bar{\theta}_{2,1}) = 6.10 \times 10^{-3}$ | $S_{50-90}(\bar{\theta}_{2,4}) = 1.30 \times 10^{-3}$ | $S_{50-90}(\bar{\theta}_{2,7}) = 2.30 \times 10^{-3}$ | 1442 |
| 4 | [90–100%] | $S_{90-100}(\bar{\theta}_{2,1}) = 3.00 \times 10^{-4}$ | $S_{90-100}(\bar{\theta}_{2,4}) = 9.25 \times 10^{-5}$ | $S_{90-100}(\bar{\theta}_{2,7}) = 3.26 \times 10^{-4}$ | 12,920 |
| - | [0–100%] | $S(\bar{\theta}_{2,1}) = 1.10 \times 10^{-3}$ | $S(\bar{\theta}_{2,4}) = 1.10 \times 10^{-3}$ | $S(\bar{\theta}_{2,7}) = 1.40 \times 10^{-3}$ | 15,001 |

Note that from expression (35) the following relation must be fulfilled:

$$\begin{aligned} S(\bar{\theta}_{2,i}) &= S_{0-100}(\bar{\theta}_{2,i}) \\ &= \tfrac{1}{N_s}[S_{0-10}(\bar{\theta}_{2,i})\cdot N_{s1} + S_{10-50}(\bar{\theta}_{2,i})\cdot N_{s2} + S_{50-90}(\bar{\theta}_{2,i})\cdot N_{s3} \\ &\quad + S_{90-100}(\bar{\theta}_{2,i})\cdot N_{s4}] \end{aligned} \quad (38)$$

Figure 20 represents graphically the information taken from Table 20, which simplifies the interpretation of results.

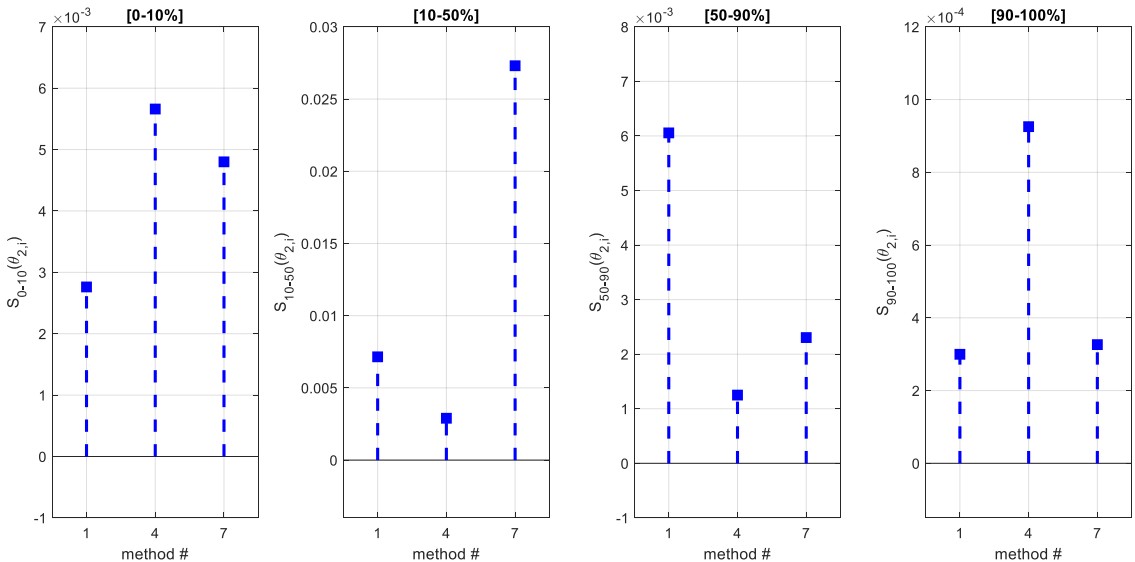

**Figure 20.** Graphical representation of data given in Table 20 for process $P_2$. From left to right the performance indexes $S_{0-10}(\bar{\theta}_{2,i})$, $S_{10-50}(\bar{\theta}_{2,i})$, $S_{50-90}(\bar{\theta}_{2,i})$, and $S_{90-100}(\bar{\theta}_{2,i})$, are shown for methods #1, #4, and #7, respectively.

Information about the accuracy of the model obtained at the different intervals of the reaction curve can be extracted from Table 20. Specifically, it can be seen that method #1 provides a model that fits the reaction curve very well in the intervals [0–10%], [10–50%] and [90–100%], while the fit is worse in the interval [50–90%], as can also be seen in Figure 15a. This method provides the best results of the three compared methods in the intervals [0–10%] and [90–100%].

The model for method #4 fits very well in the intervals [10–50%] and [50–90%], where it presents the lowest values of the three compared methods. In contrast, the value of S increases outside these intervals, as can also be seen in Figure 16b.

The model proposed by Tavakoli-Kakhki provides a good fit in the intervals [50–90%] and [90–100%], while the results for the rest of intervals are worse, particularly for [10–50%].

*4.3. Example 3*

The model (32), which has been proposed in [45] as part of a collection of systems that are suitable for testing PID controllers, is considered in this example.

Physically, this transfer function describes how temperature evolves over time in a solid medium, which assumes that the temperature is distributed in only one spatial coordinate and the heat is transferred in the direction in which the temperature decreases. This ideal transfer function contains the irrational operator $\sqrt{s}$ which may cause difficulties in a further analytical analysis.

In general, temperature control is a widespread application in the process industry, and, in particular, thermal conduction is a very common dynamic in process control.

Traditionally, the FOPDT model has been utilized to approximate this dynamic; however, it has insufficient accuracy. Since a growing body of evidence has suggested that fractional-order models are able to describe dynamic processes with higher accuracy, the proposed identification method will be used to more accurately model this process, which exhibits fractional behavior.

In order to evaluate the effectiveness of the proposed method, the estimated models for the symmetrical and asymmetrical cases will be compared with the optimal FOPDT and FFOPDT models.

Following the same procedure used in previous examples, the following FFOPDT model parameters $\overline{\theta}_{3,i} = \{K_{3,i}, T_{3,i}, L_{3,i}, \alpha_{3,i}\}$ for i = 1, . . . , 6, have been obtained in Table 21 for the different sets of points, i.e., (5-50-95%), (10-50-90%), (25-50-75%), (10-55-90%), (20-60-95%), and (20-75-95%), respectively. Model parameters for optimal FFOPDT and FOPDT models have also been calculated. Process (32) has also been approximated by the optimal FFOPDT and FOPDT models, where model parameters $\overline{\theta}_{3,i}$ for i = 7 and 8, respectively, are given in Table 21. Note that the parameters obtained in the optimal models are those that minimize the function $S(\overline{\theta})$ (33). Since the Taylor series expansion for the irrational transfer function (32) give a polynomial of half-order integrators $s^{0.5}$, $\alpha_{3,7}$ has been considered to be 0.5.

**Table 21.** Fractional-order model settings for the symmetrical and asymmetrical sets of points (5-50-95%), (10-50-90%), (25-50-75%), (10-55-90%), (20-60-95%), and (20-75-95%), respectively, and optimal FFOPDT and FOPDT model parameters.

| Method #1: (5-50-95%) | Method #2: (10-50-90%) | Method #3: (25-50-75%) | Method #4: (10-55-90%) | Method #5: (20-60-95%) | Method #6: (20-75-95%) | #7: FFOPDT Optimal | #8: FOPDT Optimal |
|---|---|---|---|---|---|---|---|
| $K_{3,1} = 1.00$ | $K_{3,2} = 1.00$ | $K_{3,3} = 1.00$ | $K_{3,4} = 1.00$ | $K_{3,5} = 1.00$ | $K_{3,6} = 1.00$ | $K_{3,7} = 1.02$ | $K_{3,8} = 0.91$ |
| $T_{3,1} = 1.26$ s | $T_{3,2} = 1.26$ s | $T_{3,3} = 1.22$ s | $T_{3,4} = 1.22$ s | $T_{3,5} = 1.13$ s | $T_{3,6} = 1.08$ s | $T_{3,7} = 1.23$ s | $T_{3,8} = 2.24$ s |
| $L_{3,1} = 0.00$ s | $L_{3,2} = 0.60$ s | $L_{3,3} = 0.22$ s | $L_{3,4} = 0.67$ s | $L_{3,5} = 0.00$ s | $L_{3,6} = 0.00$ s | $L_{3,7} = 0.0001$ s | $L_{3,8} = 0.00$ s |
| $\alpha_{3,1} = 0.5368$ | $\alpha_{3,2} = 0.5459$ | $\alpha_{3,3} = 0.5598$ | $\alpha_{3,4} = 0.5401$ | $\alpha_{3,5} = 0.5206$ | $\alpha_{3,1} = 0.5140$ | $\alpha_{3,7} = 0.5000$ | - |

In this example, an ideal thermal conduction process model has been used to demonstrate the effectiveness of the proposed identification procedure in the task of estimating the FFOPDT model parameters, compared to the optimal FOPDT model, which is traditionally used to model this type of process. Also, the results are compared with the optimal FFOPDT model to verify that with the proposed method it is possible to obtain comparable results to the latter with much less computational effort.

The step responses of the considered approximated models in Table 21 are compared with the process reaction curve and illustrated in Figures 21–24.

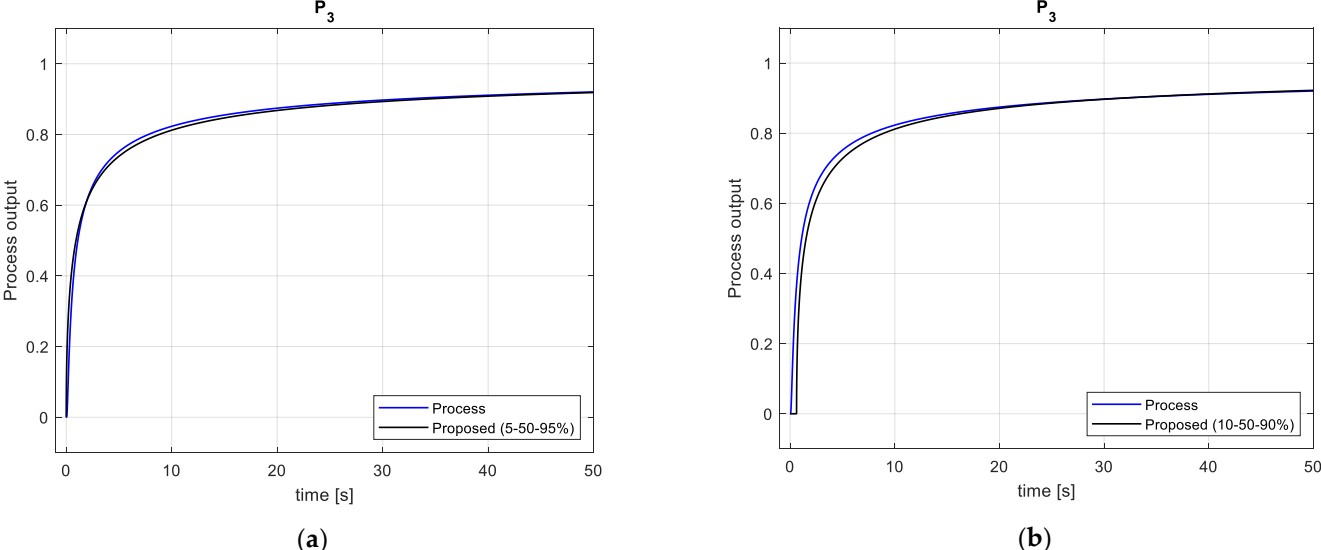

**Figure 21.** FFOPDT model step response using the proposed identification method for process P$_3$ and process reaction curve: (**a**) Symmetrical set of points (5-50-95%); (**b**) Symmetrical set of points (10-50-90%).

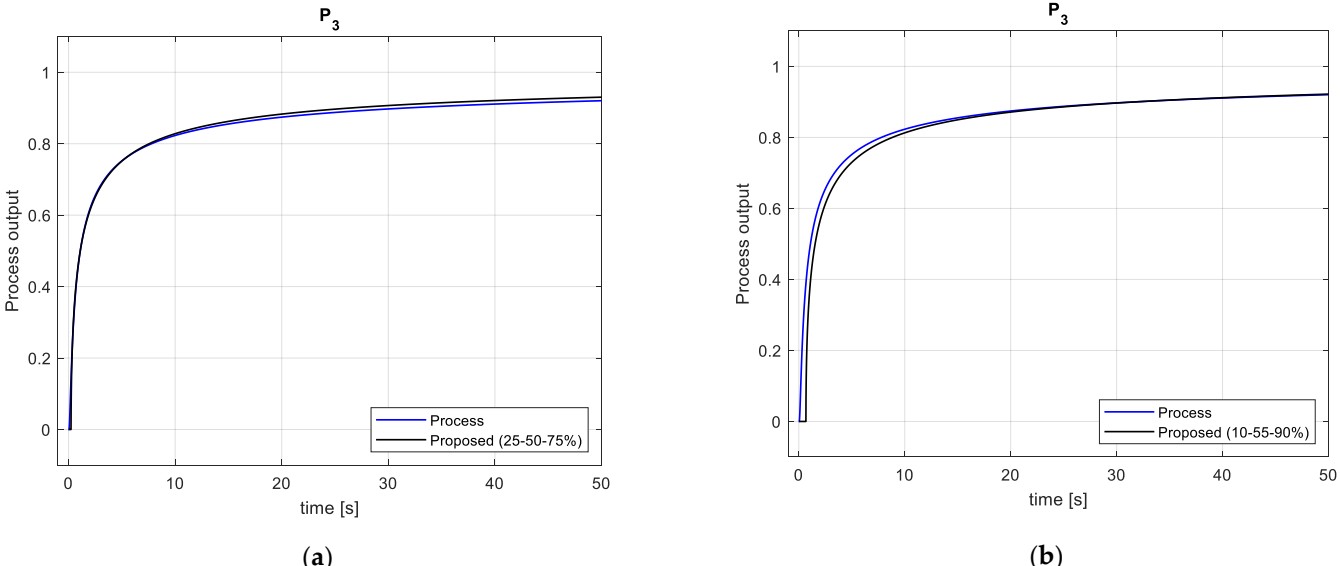

**Figure 22.** FFOPDT model step response using the proposed identification method for process P$_3$ and process reaction curve: (**a**) Symmetrical set of points (25-50-75%); (**b**) Asymmetrical set of points (10-55-90%).

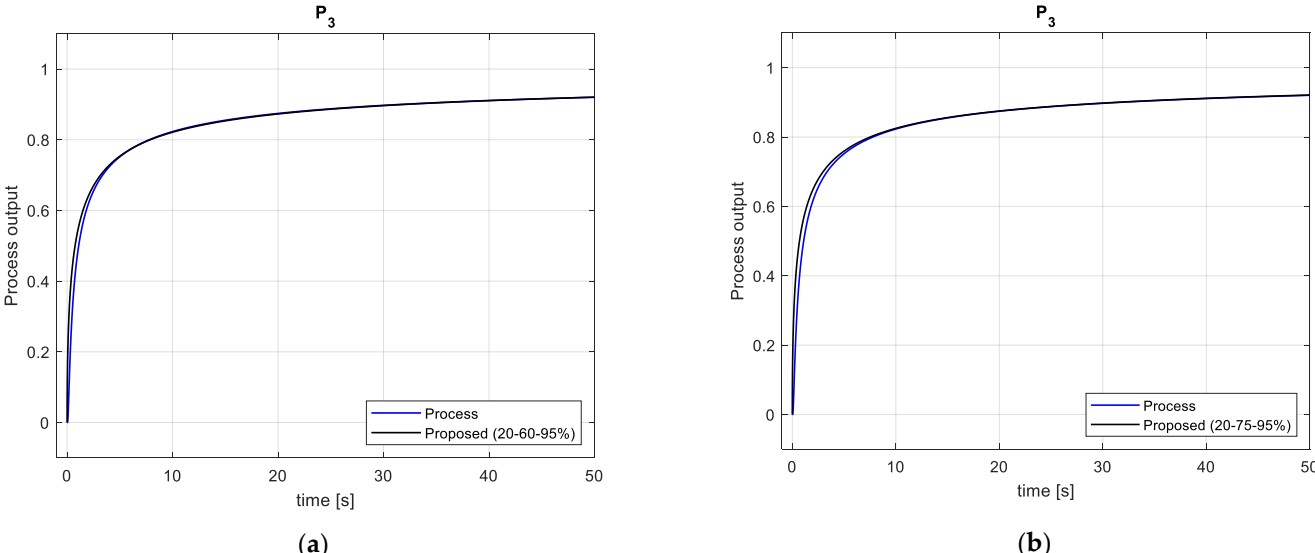

**Figure 23.** FFOPDT model step response using the proposed identification method for process P$_3$ and process reaction curve: (**a**) Asymmetrical set of points (20-60-95%); (**b**) Asymmetrical set of points (20-75-95%).

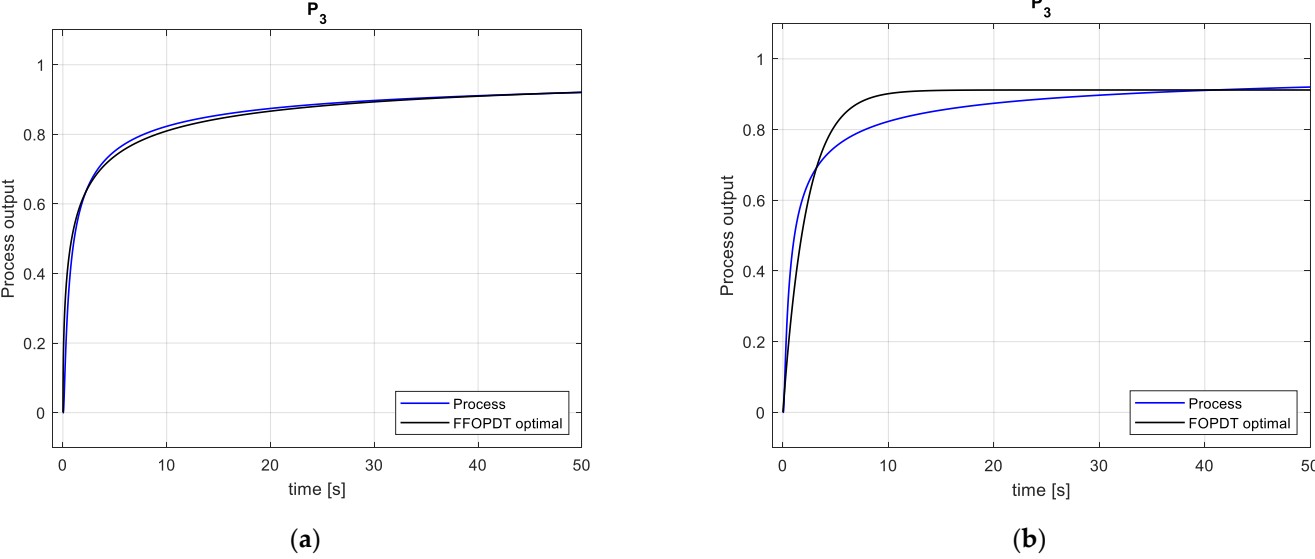

**Figure 24.** FFOPDT model step response using the proposed identification method for process P$_2$ and process reaction curve: (**a**) Optimal FFOPDT; (**b**) Optimal FOPDT.

Figures 21–23 illustrate that the models obtained with the proposed method, for both the symmetrical and asymmetrical case, provide a good fit with the process reaction curve compared to the results obtained using the optimal FFOPDT, which is shown in Figure 24a. It can also be observed that the proposed method outperforms the results obtained using the optimal FOPDT model, which is shown in Figure 24b.

Table 22 shows the values of the time-domain performance indexes $S(\overline{\theta}_{3,i})$ and $E(\overline{\theta}_{3,i})$ for process P$_3$, for the different models considered in this example. In this table, i = 1, . . . , 6 represents the different sets of points considered in the proposed identification method, and i = 7 and 8 represents optimal FFOPDT and FOPDT models, respectively.

**Table 22.** Comparison between the performance indexes obtained for the proposed method with different sets of points, symmetrical and asymmetrical, and the ones for optimal methods for FFOPDT and FOPDT models. The number of samples $N_S$ is also displayed.

| i | Method | Set of Points | $\overline{S}(\theta_{3,i})$ | $\overline{E}(\theta_{3,i})$ |
|---|--------|---------------|------------------------------|------------------------------|
| 1 | FFOPDT Proposed #1 | (5-50-95%) | $1.00 \times 10^{-4}$ | $3.00 \times 10^{-3}$ |
| 2 | FFOPDT Proposed #2 | (10-50-90%) | $3.60 \times 10^{-4}$ | $5.70 \times 10^{-3}$ |
| 3 | FFOPDT Proposed #3 | (25-50-75%) | $8.97 \times 10^{-5}$ | $8.90 \times 10^{-3}$ |
| 4 | FFOPDT Proposed #4 | (10-55-90%) | $4.19 \times 10^{-4}$ | $5.40 \times 10^{-3}$ |
| 5 | FFOPDT Proposed #5 | (20-60-95%) | $1.41 \times 10^{-4}$ | $2.00 \times 10^{-3}$ |
| 6 | FFOPDT Proposed #6 | (20-75-95%) | $1.73 \times 10^{-4}$ | $2.50 \times 10^{-3}$ |
| 7 | FFOPDT optimal | - | $1.47 \times 10^{-4}$ | $6.40 \times 10^{-3}$ |
| 8 | FOPDT optimal | - | $1.40 \times 10^{-3}$ | $3.08 \times 10^{-2}$ |
| | Number of samples | | $N_S = 15{,}001$ | $N_S = 15{,}001$ |

Table 22 shows that methods #1, #5, and #6 provide E values comparable to that obtained for the optimal FFOPDT model. On the other hand, it is also observed that the FOPDT model approximates the process dynamics with insufficient accuracy. In fact, for example, methods #1, #5, and #6 have E values that are 10.3, 15.4, and 12.3 times lower than that obtained by the optimal FOPDT model.

## 5. Experimental Results

One of the objectives of this paper is to verify the applicability of the proposed identification procedure in a laboratory prototype and to obtain insight into the practical issues related to its implementation on industrial control hardware.

In this section, a thermal process-based hardware-in-the-loop experimental setup will be used to validate the effectiveness of the proposed identification procedure implemented on microprocessor-based control hardware.

This section is organized as follows. Firstly, the laboratory prototype to be used is described. After that, the controlled process is configured to test the identification procedures proposed in this work for both the symmetrical and the asymmetrical case. Since the prototype can be set up in three different configurations, this section describes the selected configuration and defines the different components and variables that make up the controlled process. Then, the hardware control architecture to be used for the operation of the prototype and the practical implementation of the model estimation procedures are briefly presented. Next, the identification procedures are applied to the thermal laboratory prototype using a microprocessor-based control hardware, confirming the applicability of these procedures in industrial equipment. This constitutes one of the main results of this paper. Finally, some remarks and final comments about practical issues in the microprocessor-based implementation are offered in this context.

### 5.1. Description of the Prototype

Recently, a thermal-based experimental setup has been designed and built in the Faculty of Engineering, University of Deusto. The considered prototype consists of two clearly different parts, as can be seen in the 3D-model layout in Figure 25.

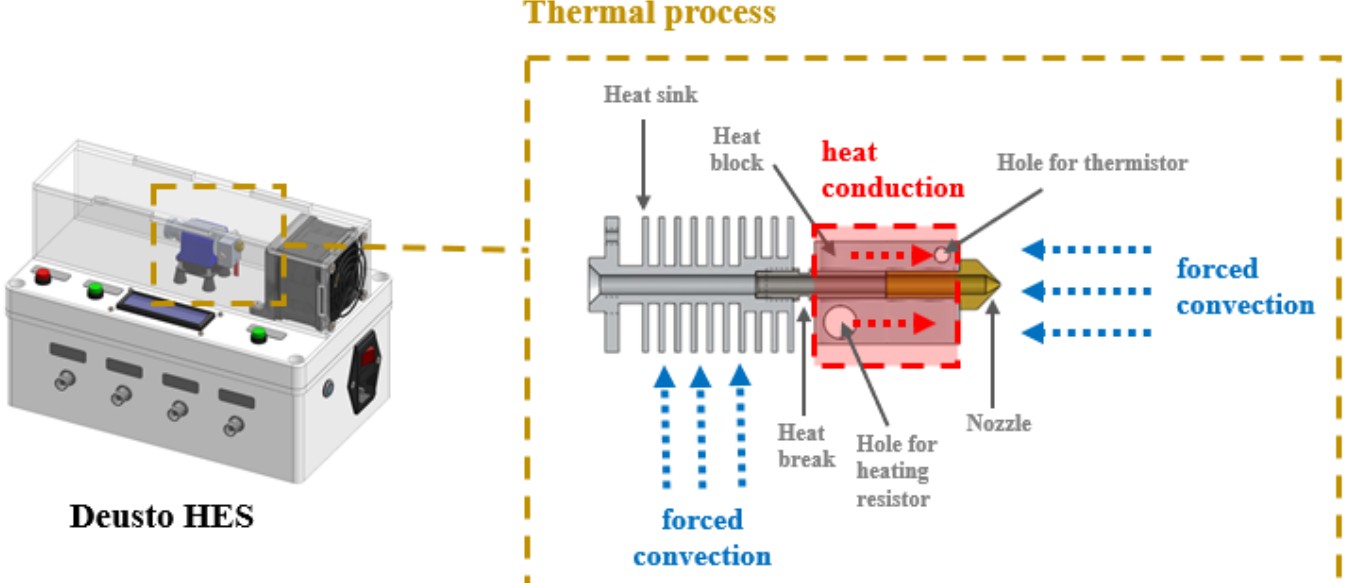

**Figure 25.** Deusto Heater Experimental Setup—3D-model layout of the prototype and schematic detail of the thermal process.

- The upper part of the equipment, where there is a methacrylate duct, the head of a 3D-printer extruder, and an air fan in front of the hot end. User LEDs, an LCD display, a user button, and four BNC output connectors to display the main process variables on an oscilloscope, can be found outside of the enclosure.
- The inner part of the enclosure, where the power supply and all the hardware and electrical components necessary for the correct operation of the experimental setup can be found. The connection of the input and output signals to the control hardware has also been arranged through a standard 34-way IDC connector, which is placed on one side of the box.

A detailed description of this laboratory equipment, which is named Deusto Heater Experimental Setup (Deusto HES), can be found in [46].

*5.2. Reconfigurable Controlled Process*

The thermal process considered in this paper takes place on the upper part of the prototype. The equipment brings about the thermodynamic process of temperature control in a 3D-printer extruder head.

It should be noted that the head of the 3D-printer is not a conventional component, but the dimensions of the heat block have been conveniently modified. The head of the 3D-printer is just used as a heating element (final controlling element) and there is no type of extrusion process, as takes place in a real 3D-printer.

The control objective in this process is to achieve a temperature in the heat block $T(t)$ that reaches the value set by the user $T^{SET}$, despite the disturbances that may take part in the process.

If it is assumed that the physical properties do not vary, only the effects of the heating power and the air fan on the controlled variable $T(t)$ should be considered, as shown in Figure 25. Accordingly, from a thermodynamical point of view, the only source of heat in the system is represented by the heat conduction originated in the resistor hole, which exhibits a fractional behavior [47]. On the other hand, another phenomenon that takes place in the process is the forced convection on the heat block generated by the airflow due to the air fan [48,49]. As both phenomena can be controlled by the control hardware, from a control point of view the controlled process configuration can be set using software components, as will be discussed below.

In this paper, the heating element is used as the final control element while keeping constant the speed of the air fan. Figure 26 represents the block diagram of the controlled process in the selected configuration, while the different components, variables, and units are listed in Table 23. However, two additional configurations are available with this equipment, constituting a reconfigurable controlled process, as is explained in detail in [50].

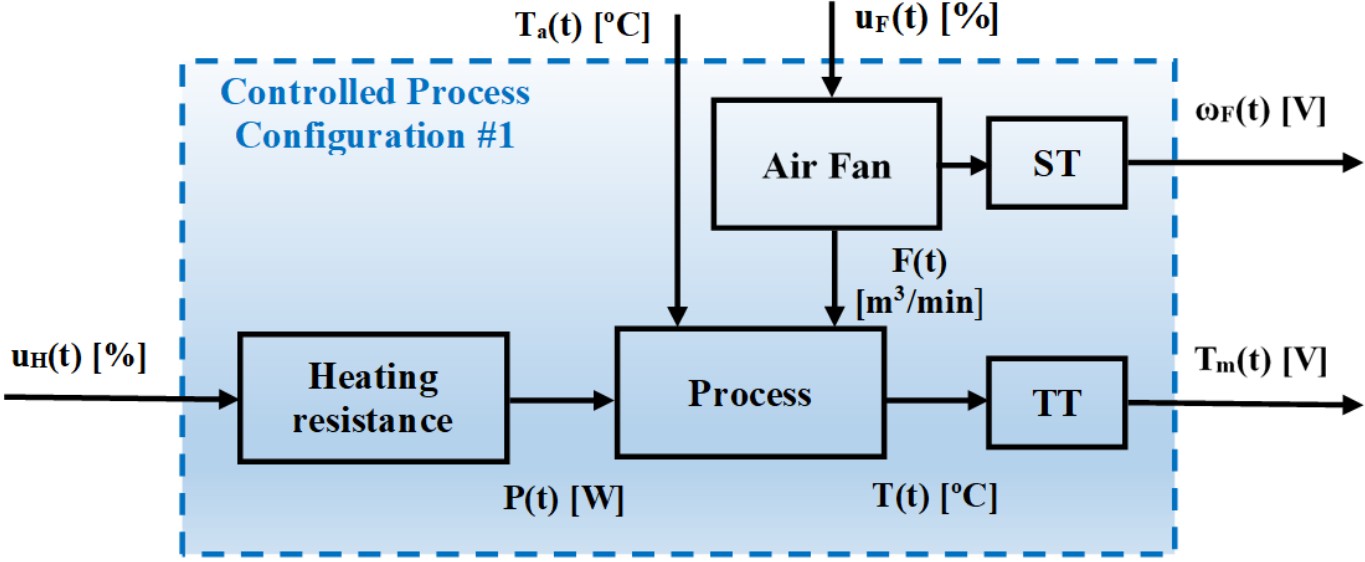

**Figure 26.** Block diagram for the considered controlled process configuration.

**Table 23.** Main process variables and components in the considered controlled process configuration.

| Process Variables or Components | Controlled Process Configuration #1 |
|---|---|
| Controlled variable | Temperature in the heat block T(t) [°C] |
| Manipulated variable | Power delivered to the heat block by the heating resistance P(t) [W] |
| Measured variables | Temperature measured by the thermistor $T_m$(t) [V] |
| | Rotational speed of fan $\omega_F$(t) [V] |
| Control signal | Output of the controller $u_H$(t) [%] |
| Final control element | Heating resistance |
| Measurement devices | Temperature transmitter (TT) and Frequency transmitter (ST) |
| Disturbances | Ambient temperature $T_a$(t) [°C] and command signal to air fan $u_F$(t) [%] |

*5.3. Control Hardware*

In this paper, NI myRIO-1900 equipment was used as a control hardware device, although any other type of microprocessor or hardware device could be easily incorporated.

Figure 27 illustrates the scheme of the hardware architecture used for the operation of this prototype and for the practical implementation of control and model estimation algorithms. This hardware architecture has been proposed in [50] and the use of this control hardware for the implementation of integer- and fractional-order PID controllers is explained in detail. In particular, the design and experimental validation of the fractional-order controllers implemented in several control technologies were applied to this thermal-based experimental setup in order to demonstrate the effectiveness of the proposed hardware architecture.

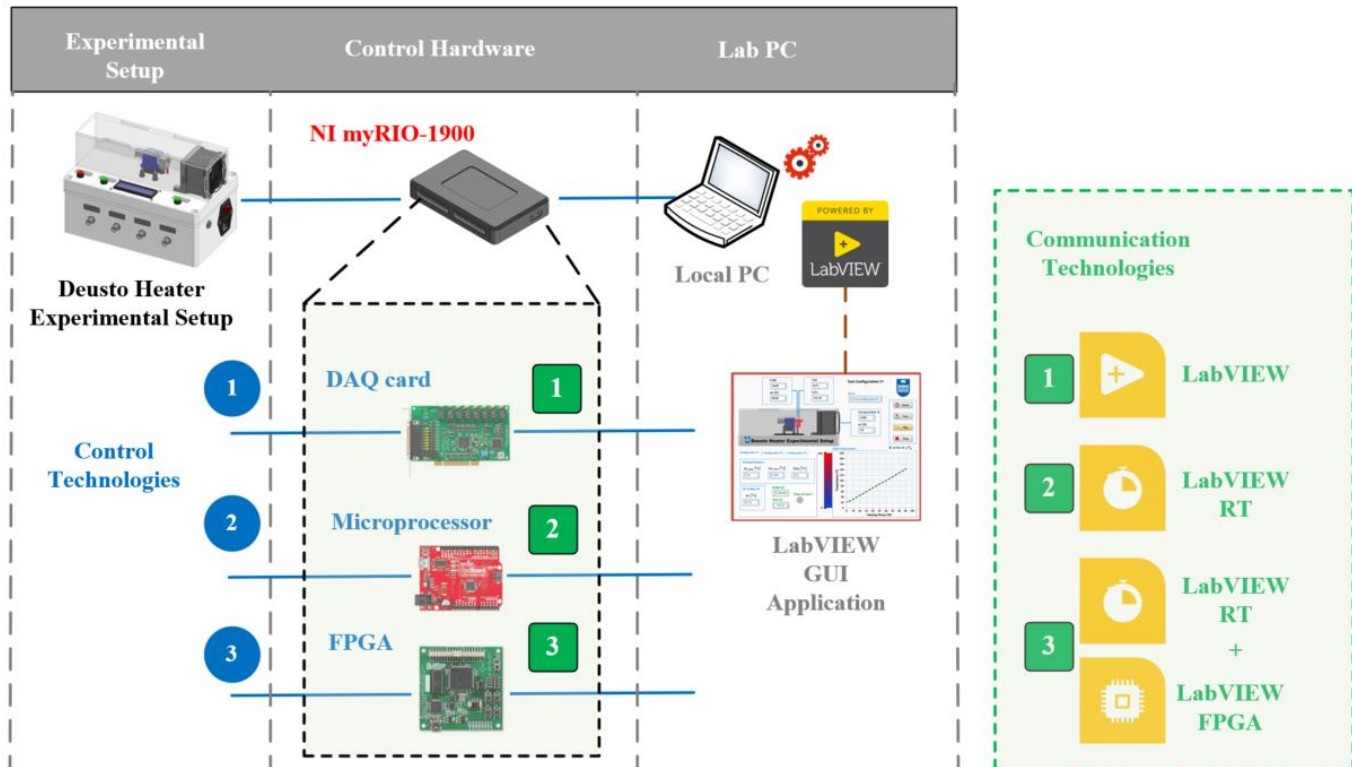

**Figure 27.** Scheme of the hardware architecture used for operation and implementation of fractional-order model identification algorithms applied to the laboratory prototype.

The flexibility required in the industrial context is offered by this control hardware and the configuration of the proposed control architecture provides the following technologies for the practical implementation of model identification procedures:

1. DAQ mode;
2. Microprocessor-based mode;
3. FPGA-based mode.

LabVIEW has been used as a programming language in this prototype. LabVIEW, which has become a de facto standard in industry, especially in the areas of industrial measurement and control, with the latest and most advanced programming techniques and transparency in the access to hardware devices, facilitates not only the application of different modelling techniques, but also the practical implementation of conventional and advanced control algorithms.

Depending on the control technology that is being used, the communication technologies between the control hardware and the local or remote PC are programmed with the following software components, as indicated in Figure 27 and detailed in [46].

1. LabVIEW;
2. LabVIEW RT;
3. LabVIEW RT and LabVIEW FPGA

Without loss of generality, in this paper only the microprocessor-based hardware mode has been used to operate the prototype and estimate fractional-order model parameters applying the proposed identification procedure.

### 5.4. Model Estimation

For the purposes of this paper, it is enough to deal with the static and dynamic characteristics of the controlled process, estimating an FFOPDT model from the process

reaction curve at a certain operating point, as will be considered in this section. To do this, a LabVIEW-based application has been implemented.

Consider now that an open-loop step-test experiment is applied to the aforementioned thermal laboratory setup with the controlled process configuration detailed in Figure 26.

Initially, the fan and the control signal to the heating element are at $u_F$ = 10% and $u_H$ = 30%, respectively.

At instant t = 0 s, a step change of amplitude $\Delta u_H$ = 30% is applied, from $u_H$ = 30% to 60%, while the value of $u_F$ is kept constant, as shown in Figure 28a. The measured temperature in the heat block, which ranges from 60.5 to 102.5 °C ($\Delta T_m$ = 42 °C), is recorded in Figure 28b. This output is a noisy signal and that noise is reduced with a first-order low-pass filter that usually is used for filtering purposes. The sample time used in this experiment was $T_S$ = 100 ms.

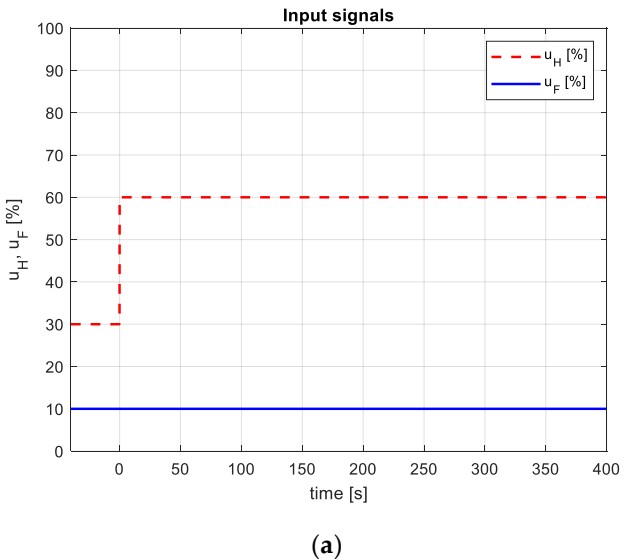

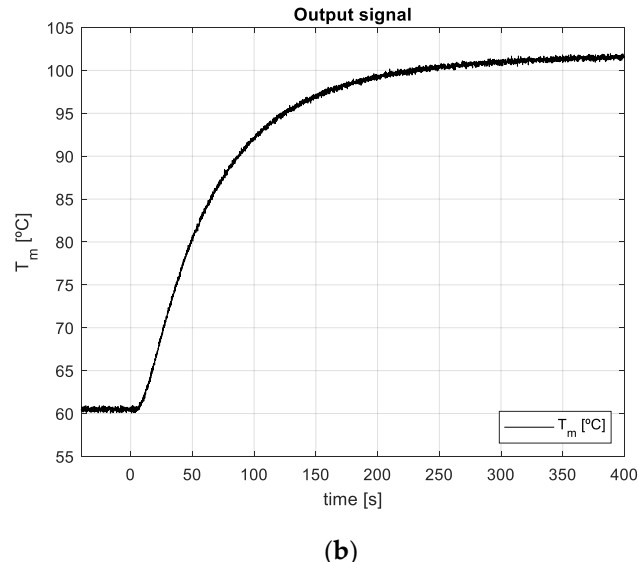

(**a**)  (**b**)

**Figure 28.** Experimental open-loop step test for determining FFOPDT model parameters: (**a**) Control signal $u_H(t)$ [%] and command signal to air fan $u_F(t)$ [%]; (**b**) Process reaction curve $T_m(t)$ [°C] obtained from an open-loop step-test experiment.

From the filtered output signal shown in Figure 28b, the process information in Table 24 is obtained.

**Table 24.** Process information for methods #2 and #4, respectively, collected from the process reaction curve for fractional-order model identification of the thermal process.

| Symmetrical | Asymmetrical |
|---|---|
| **Method #2:**<br>**(10-50-90%)** | **Method #4:**<br>**(10-55-90%)** |
| $\Delta u = \Delta u_H = 30\%$ | |
| $\Delta y = \Delta T_m = 42\ °C$ | |
| $t_{10} = 16.8000\ s$ | |
| $t_{50} = 53.3000\ s$ | $t_{55} = 59.9000\ s$ |
| $t_{90} = 174.5000\ s$ | |

Table 25 contains the different model parameters for the thermal process at the considered operating point obtained using the identification methods #2 and #4 for FFOPDT models, using methods proposed by Alfaro in [11] and Vitecková et al. in [14] for FOPDT

models, and using methods proposed by Stark in [16] and Jahanmiri and Fallahi in [15] for SOPDT models, respectively.

**Table 25.** Model parameters for the thermal process at the considered operating point obtained using the identification methods #2 and #4 for FFOPDT models, using methods proposed by Alfaro and Vitecková for FOPDT models, and using methods proposed by Stark and Jahanmiri and Fallahi for SOPDT models, respectively.

| Method #2 (10-50-90%) | Method #4 (10-55-90%) | FOPDT [11] (25–75%) | FOPDT [14] (33–70%) | SOPDT [16] (15-45-75%) | SOPDT [15] (2-70-90%) |
|---|---|---|---|---|---|
| $K_{4,1} = 1.40\,°C/\%$ | $K_{4,2} = 1.40\,°C/\%$ | $K_{4,3} = 1.40\,°C/\%$ | $K_{4,4} = 1.40\,°C/\%$ | $K_{4,5} = 1.40\,°C/\%$ | $K_{4,6} = 1.40\,°C/\%$ |
| $T_{4,1} = 49.52\,s$ | $T_{4,2} = 48.43\,s$ | $T_{4,3} = 64.26\,s$ | $T_{3,4} = 63.37\,s$ | $T_{4a,5} = 59.92\,s$ | $T_{3,4} = 70.92\,s$ |
| $L_{4,1} = 11.51\,s$ | $L_{4,2} = 11.64\,s$ | $L_{4,3} = 10.20\,s$ | $L_{3,4} = 10.25\,s$ | $T_{4b,5} = 13.68\,s$ | $T_{3,4} = 0.092\,s$ |
| $\alpha_{4,1} = 0.9462$ | $\alpha_{4,2} = 0.9430$ | - | - | $L_{4,5} = 0.00\,s$ | $L_{4,6} = 9.10\,s$ |

In this example, the thermal-based experimental setup has been utilized in order to illustrate the effectivity and applicability of the proposed identification method in an industrial environment.

The step responses of the estimated models for the symmetrical (10-50-90%) and asymmetrical (10-55-90%) case, and the ones for FOPDT and SOPDT models obtained using the considered classical methods, are compared with the process reaction curve and illustrated in Figures 29–31, respectively. The corresponding representative points are also displayed in these figures.

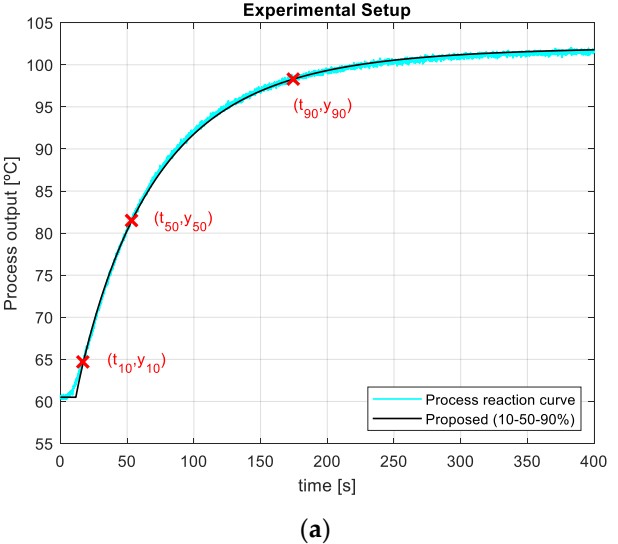
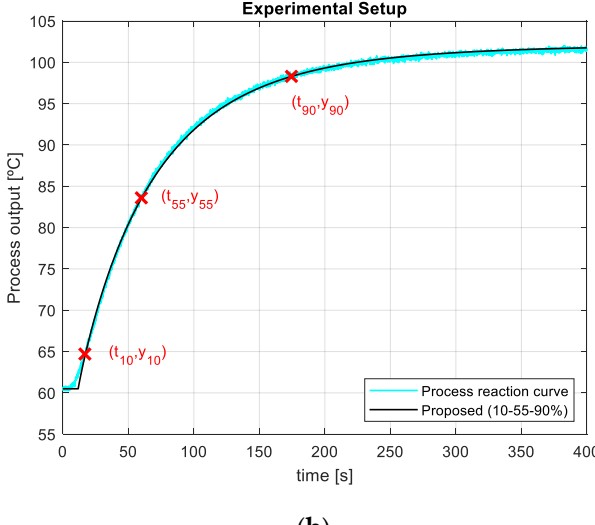

(**a**)    (**b**)

**Figure 29.** FFOPDT model step response using the proposed identification method for the experimental setup in a certain operating point and process reaction curve: (**a**) Symmetrical set of points (10-50-90%); (**b**) Asymmetrical set of points (10-55-90%).

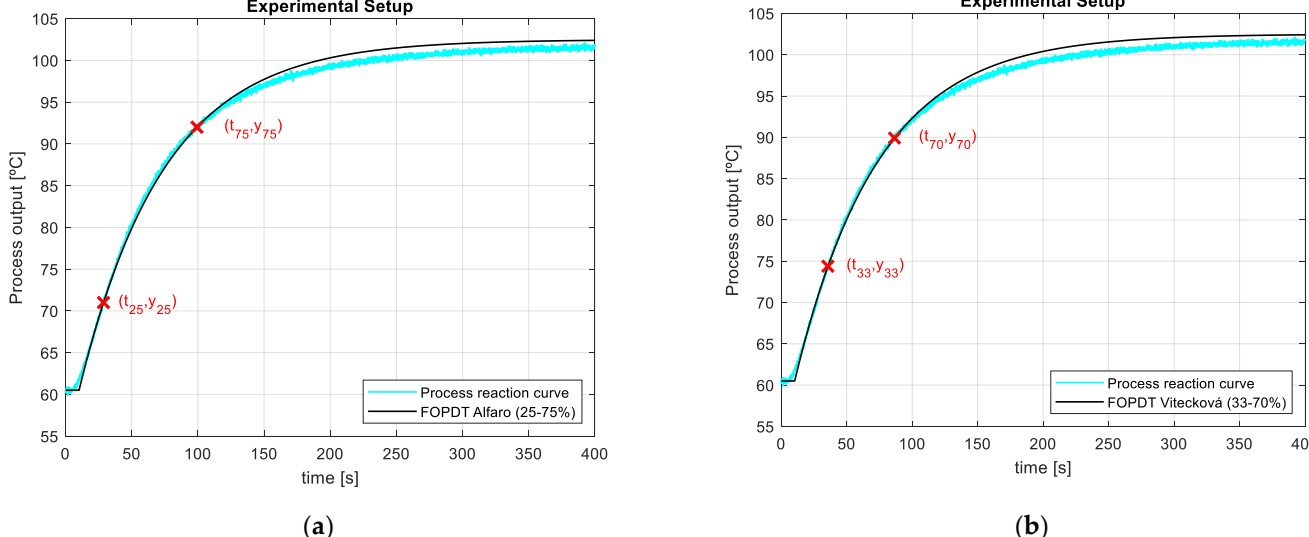

**Figure 30.** FOPDT model step response using two-point identification methods for the experimental setup in a certain operating point and process reaction curve: (**a**) Method proposed by Alfaro in [11] with symmetrical set of points (25–75%); (**b**) Method proposed by Viteckóva in [14] with asymmetrical set of points (33–70%).

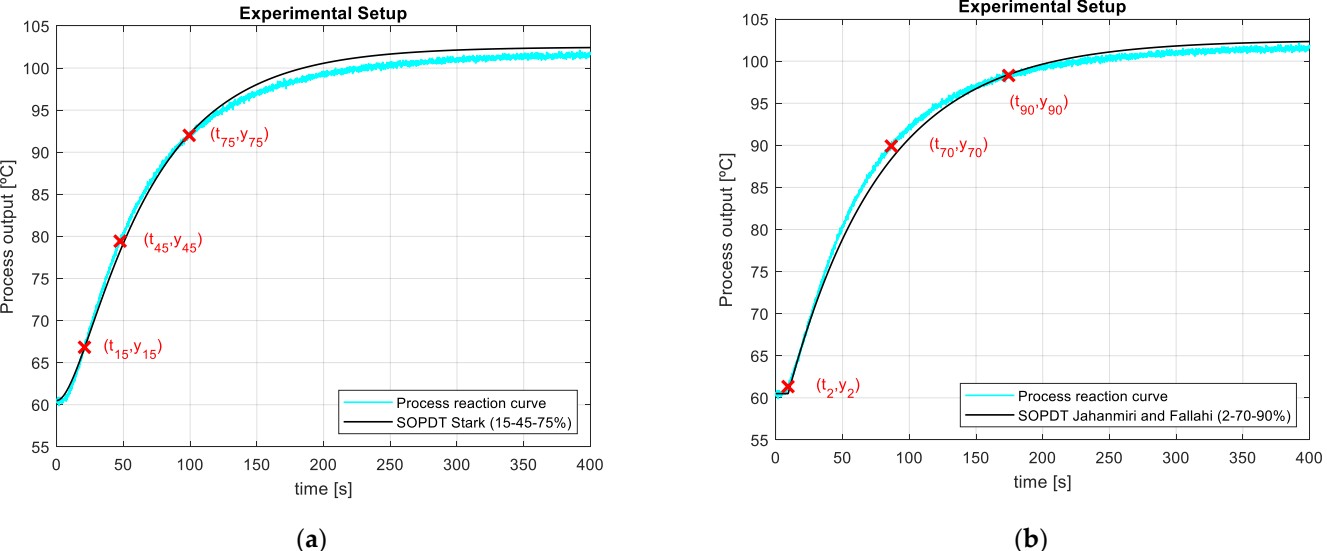

**Figure 31.** SOPDT model step response using three-point identification methods for the experimental setup in a certain operating point and process reaction curve: (**a**) Method proposed by Stark [16] with symmetrical set of points (15-45-75%); (**b**) Method proposed by Jahanmiri and Fallahi [15] with asymmetrical set of points (5-70-90%).

Figure 29 illustrates that the FFOPDT models obtained using methods #2 and #4 give a good fit to the thermal process reaction curve in comparison to the results obtained for the FOPDT and SOPDT models, which are shown in Figures 30 and 31, respectively. Note that the computational effort of all the methods considered in this experimental example is similar since all are analytical methods based on the process reaction curve.

Table 26 shows the values of the time-domain performance indexes $S(\overline{\theta}_{4,i})$ and $E(\overline{\theta}_{4,i})$ for the different approximated models considered in this example.

**Table 26.** Comparison of the approximated models obtained with methods #2 and #4 and models for FOPDT and SOPDT in terms of time-domain model performance indices for the thermal-based process at a certain operating point. The number of samples $N_S$ is also displayed.

| i | Identification Method | Set of Points | $\overline{S}(\theta_{4,i})$ | $\overline{E}(\theta_{4,i})$ |
|---|---|---|---|---|
| 1 | FFOPDT Proposed method #2 | (10-50-90%) | $7.59 \times 10^{-5}$ | $6.90 \times 10^{-3}$ |
| 2 | FFOPDT Proposed method #4 | (10-55-90%) | $6.62 \times 10^{-5}$ | $6.00 \times 10^{-3}$ |
| 3 | FOPDT Alfaro | (25-75%) | $7.69 \times 10^{-4}$ | $2.50 \times 10^{-2}$ |
| 4 | FOPDT Vitecková | (33-70%) | $8.47 \times 10^{-4}$ | $2.59 \times 10^{-2}$ |
| 5 | SOPDT Stark | (15-45-75%) | $1.20 \times 10^{-3}$ | $3.20 \times 10^{-2}$ |
| 6 | SOPDT Jahanmiri and Fallahi | (2-70-90%) | $8.35 \times 10^{-4}$ | $2.51 \times 10^{-2}$ |
| | Number of samples | | $N_S = 4001$ | $N_S = 4001$ |

In this table, one can observe that the result obtained by method #4 is slightly better in terms of E than the one obtained using method #2. The results obtained by fractional-order identification methods outperform those obtained using integer-order models. Specifically, the E values obtained by using methods #2 and #4 are 3.6, 3.7, 4.6, and 3.6 times lower than those obtained by using the methods of Alfaro, Vitecková, Stark, and Jahanmiri and Fallahi, respectively.

LabVIEW-Based Implementation

As discussed previously, a LabVIEW-based application has been developed to operate the prototype and implement the proposed identification procedure, demonstrating its applicability on a microprocessor-based hardware device.

Figure 32 illustrates the front panel of the LabVIEW-based application for fractional-order model estimation using the proposed identification procedure, considering both the symmetrical and the asymmetrical case.

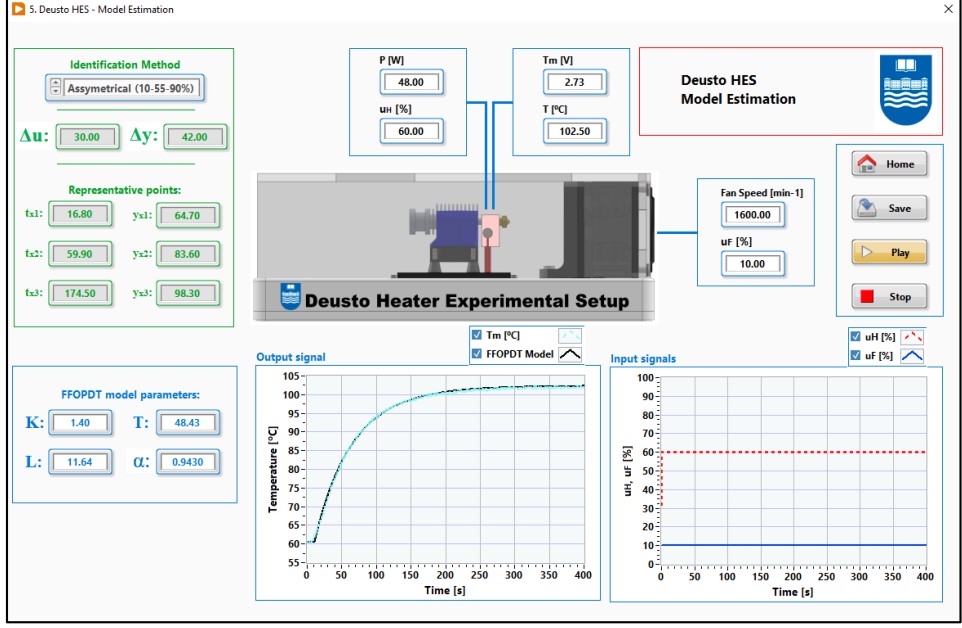

**Figure 32.** Front panel of the LabVIEW-based application for implementing the fractional-order identification algorithm proposed in this paper. This figure illustrates the different features available in this application.

Once the identification method has been selected, and according to the identification procedure, the variation in input signal $\Delta u$ and process output $\Delta y$, and times required to reach $x_1\%$ ($t_{x1}$), $x_2\%$ ($t_{x2}$), and $x_3\%$ ($t_{x3}$) of the total process output change on the reaction curve, must be taken in order to obtain FFOPDT model parameters $\theta_P = \{K, T, L, \alpha\}$.

The algorithm illustrated in Algorithm 1 has been developed in order to simplify software implementation of the identification procedure proposed in this paper.

---

**Algorithm 1. Three-Point Identification Method for FFOPDT Models**

---

**Input**: Selected symmetrical or asymmetrical method and $\{\Delta u, \Delta y, t_{x1}, t_{x2}, t_{x3}\}$ collected from the process reaction curve

---

**Output**: Fractional-order model parameters $\theta_P = \{K, T, L, \alpha\}$

---

**1**: Select the corresponding symmetrical or asymmetrical method.
**2**: Collect process data $\{\Delta u, \Delta y, t_{x1}, t_{x2}, t_{x3}\}$ from the process reaction curve.
**3**: Obtain the process gain K using (18).
**4**: Obtain the value of the times ratio $\Delta$ by using Equation (21).
**5**: Obtain the value of $\alpha = f_1(\Delta)$ by using Equation (27).
**6**: Obtain the value of functions $f_2(\alpha)$ and $f_3(\alpha)$ by using Equations (28) and (29), respectively.
**7**: Calculate the value of T using Equation (22).
**8**: Calculate the value of L using Equation (23).

---

This application presents the following features, which can be observed in Figure 32:

1.  Selection of the identification procedure, which presents the following options:
    a.  Symmetrical case (x-50-(100−x)%): (5-50-95%), (10-50-90%), or (25-50-75%).
    b.  Asymmetrical case ($x_1$-$x_2$-$x_3$%): (10-55-90%), (20-60-95%), or (20-75-95%).
2.  Determination of process data $\{\Delta u, \Delta y, t_{x1}, t_{x2}, t_{x3}\}$ from the process reaction curve.
3.  Estimation of the FFOPDT model parameters: $\theta_P = \{K, T, L, \alpha\}$.
4.  Graphs for registering control signal $u_H(t)$ [%], command signal to air fan $u_F(t)$ [%], process reaction curve $T_m(t)$ [°C], representative points of the process reaction curve $\{(t_{x1}, y_\alpha(t_{x1})), (t_{x2}, y_\alpha(t_{x2})), (t_{x3}, y_\alpha(t_{x3}))\}$, and step response of the identified model.
5.  Export the experimental data in Excel or text-format.

*5.5. Remarks and Final Comments*

Practical application of the proposed fractional-order model identification algorithm on laboratory equipment allows gaining experience about practical issues related to its implementation on industrial control hardware. Some remarks and comments on industrial practice in this context are discussed below.

**Remark 1.** *Measurement noise. The main disadvantage provided by the use of feedback is that measurement noise is injected into the loop. Noise generally generates undesirable motion of the final controlling elements, which may cause wear and possible breakdown.*

It is common practice that open-loop model identification procedures use process information based on a noise-free process reaction curve; see, e.g., [5] for integer-order systems, and [31] for fractional-order systems. However, it is usual in an industrial environment that the controlled process feedback signal includes measurement noise that must be properly filtered for model identification and control purposes.

It is important to take into account that the filter dynamics will be an integral part of the controlled process to be identified. As this will influence and add a lag to the loop dynamics, a measurement filter must be set before any controlled process model identification and/or controller tuning [18].

In this context, it is very difficult to derive general conclusions about the influence of measurement noise on the identification procedure, because this will depend on the measurement noise and filter characteristics.

The results of the experiment conducted in this section serve to demonstrate the applicability of the proposed identification method on laboratory equipment and its implementation on industrial control hardware, which presents similarities with the industrial environment.

**Remark 2.** *Model uncertainty. In the industrial context, processes exhibit nonlinearities, i.e., the dynamic characteristics of the process and therefore those of the identified FFOPDT model—gain, time constant, apparent dead-time and fractional order—will change with the operating point. The operating point of a control system may vary due to a change in the setpoint or as a result of the effect of disturbances.*

Therefore, it must be considered that there is an implicit uncertainty in the nominal model.

In general, there are two approaches in the technical literature when considering the parametric uncertainty of the plant in an identification method; see, e.g., [39].

- The first approach is to incorporate the uncertainty explicitly into the model. This typically makes the identification procedure more complicated.
- The second approach consists of taking into account the potential changes in the controlled process dynamics and model uncertainties in the design phase of the controller; see, e.g., [2] for integer-order controllers and [19] for fractional-order controllers. A common application of this second approach is to ensure a certain degree of robustness of the designed control system to guarantee its stability under variations in the process characteristics.

In the context of this work, the primary use of the identified fractional-order model is for control purposes. Consequently, the approach to be used will be the second one.

**Rules of thumb in selecting sets of points:** In Sections 3.1 and 3.2 some rules of thumb, which are based on experimentation, have been provided to obtain insight about the selection of the points for the symmetric and asymmetric case, respectively, in the context of the proposed identification method.

Based on the results obtained in examples 1–3 in Section 4, a recommendation on the selection of the set of points for the proposed identification method can be provided:

1. In the symmetrical case, the set of points in method #1 gives the best results in terms of the performance index E.
2. In the asymmetrical case, the set of points in methods #5 and #6 give quite similar results in terms of E, although #6 is slightly better.

**Final comments and discussion:** This section contains a discussion about the usefulness of, and the improvement obtained by, using an FFOPDT model instead of an integer-order model, with FOPDT, DPPDT, and SOPDT models being those most commonly used in the industrial context.

In this paper, an identification method for FFOPDT models has been presented. This method is focused on processes characterized by an S-shaped response (monotonic) that exhibits a fractional behavior.

There is no exact definition to describe a fractional dynamic behavior in the technical literature. According to [51], a system has a fractional behavior if its input and output are linked by a function of the form $t^{\upsilon-1}$, $\upsilon \in \mathbb{R}$, $\upsilon < 1$.

Although an implicit link exists in the literature between fractional behaviors and fractional-based models, they are two distinct concepts. One designates a property or a particular behavior of a physical system, while the other refers to a model class that can capture fractional behaviors.

Fractional behaviors appear in many physical-, biological-, or thermal-based processes, among others [19,20]. In spite of the slow dynamics, real processes exhibiting fractional behavior with values of $\alpha$ in the range proposed in this paper can be found in electrical

engineering, motion controls, and process control, with thermal-based processes being the most important type encountered in the process industry [47].

Since these kinds of behaviors are ubiquitous, the existence of methods for identification of simple-structure fractional-order models is of significant interest.

The main issue for adopting fractional-order models in industry can be summarized in the form of the following question: "Is the additional effort to consider a fractional-order model worth it, in order to obtain a more accurate model and, eventually, a better control performance?".

In the industrial context, the apparent benefit of using fractional calculus has been justified in the literature in terms of a more precise modelling; see, e.g., [20,34,36].

More specifically in the context of this work, the proposed identification method for symmetrical and asymmetrical sets of points has been compared with several two- and three-point identification methods for integer-order models, which are based on data collected from the process reaction curve. It has been illustrated that the proposed method outperforms significantly methods for integer-order models. In some cases, even better results are obtained with the estimated fractional-order model than with the optimal integer-order model.

If the process reaction curve exhibits fractional behavior, the estimated FFOPDT model will more accurately fit the reaction curve compared to the integer-order model. Note that the proposed identification method allows characterizing the existence of fractional behavior in measured data collected from the process reaction curve.

The identification procedure presented in this paper is intended for control purposes, as considered previously. In the industrial context, large process industries have hundreds or thousands of control loops. In order for such an identification procedure to have a significant impact in the industrial environment, simplicity is a fundamental feature.

In the context of this work, there is a trade-off between accuracy and computational effort. Although this identification procedure provides good results in comparison with other well-known integer- and fractional-order identification methods, it is possible to find methods that improve further the accuracy of the estimated fractional-order model at a cost of more complex algorithms or a higher computational effort. Another aspect to highlight is that the proposed method is analytical, which facilitates its applicability in terms of a lower computational effort compared to complicated identification algorithms generally based on optimization.

## 6. Conclusions

In this paper, an identification procedure for FFOPDT models, which is based on the information obtained from three points on the process reaction curve, is presented. This identification procedure has been validated for both cases, considering three symmetrical and asymmetrical points on the reaction curve.

Three simulation examples, with processes exhibiting different dynamics, have been used to verify the simplicity and effectiveness of this identification procedure for the symmetrical and asymmetrical cases. Good results have been obtained in comparison with other well-recognized integer- and fractional-order identification methods which are also based on information taken from the process reaction curve, especially when simplicity is emphasized.

A thermal process-based experimental setup has also been used, where the proposed identification procedure has been implemented in a microprocessor-based control hardware, confirming the applicability of this method in an industrial equipment.

Besides effectiveness, the main characteristic exhibited by this method is simplicity. This feature is of significant importance in the industrial context, where easy-to-implement identification methods are required.

Some comments and reflections have been offered in the context of industrial practice.

It is worth pointing out that the identification method proposed in this paper is restricted to values of the fractional order in the range $0.5 \leq \alpha \leq 1.0$, and as a future

work, this method is being extended using the same methodology for processes with underdamped step response, extending the range of the fractional order to $1.0 \le \alpha \le 2.0$.

**Author Contributions:** Conceptualization, J.J.G. and P.G.B.; methodology, J.J.G. and P.G.B.; software, J.J.G.; investigation, J.J.G.; writing—original draft preparation, J.J.G.; writing—review and editing, J.J.G. and P.G.B.; supervision, J.J.G. and P.G.B.; project administration, P.G.B.; funding acquisition, P.G.B. All authors have read and agreed to the published version of the manuscript.

**Funding:** The authors would like to thank the Basque Government for its partial funding support through the TRUSTIND ELKARTEK R&D (ref. KK-2020/00054) and REMEDY (ref. KK-2021/00091) projects.

**Institutional Review Board Statement:** Not applicable.

**Informed Consent Statement:** Not applicable.

**Data Availability Statement:** Not applicable.

**Acknowledgments:** The support of Unai Conejo as Lab Assistant in the construction of the Deusto HES prototype is acknowledged.

**Conflicts of Interest:** The authors declare no conflict of interest.

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
