# Peer review of "Proposal of a General Identification Method for Fractional-Order Processes Based on the Process Reaction Curve"

_fractalfract, doi:10.3390/fractalfract6090526_

Round 1

Reviewer 1 Report

The authors present an interesting idea about the identification method for a fractional order model, based on the process reaction curve.

The general idea presented in the paper is very interesting and with a very practical point of view. This, considering that in real industrial situations it is very common to obtain information for process from the reaction curve and to use it to identify an integer model as FOPDT, SOPDT, DPPDT that have all the characteristic to be overdamped dynamics. Note that the model identification is just an intermediate step (do not require any implementation), only to use the model to achieve de controller parameters, normally PI or PID controllers.

The introduction of fractional calculus has been increased in recent years. For control, it is still the problem how implement those kind of controllers. However, for identification its use is very powerful because can provide more information for processes and once again, I consider this is one of the main potential contributions of the paper.

From all the above, the most potential use for fractional order models (like eq 13) is the possibility to represent very different dynamics (overdamped and underdamped). Note that 1<=alpha<=2 is where the FFOPDT model has more capabilities. Processes for alpha<1 are not common in real applications.

Some comments:

- The paper can be shortened, some parts are too long and can be presented in a more simple way. For example, there are too many figures that show the same information for different cases and can be simplify using just one to illustrate the procedure.

- Delete words like "new", "novel", etc, that are not suitable for research papers.

- Even the studied range is for alpha in [0.5 1.1], all the values of the examples are lower than 1, and in the range between 0.74 and 0.98.

Some concerns:

  • From the reviewer point of view, for fractional order models, the acceptable range for alpha needs to include values for underdamped dynamics.
  • The study is very complete, however it does not provide any final recommendations about which is the final choice for the points for the identification method.
  • For validation of the proposal in the examples:
    • Use the Benchmark proposed by Astrom and Hagglud to test your method. Processes of eq 30 and 31 are not the best option. Look also for processes with different dynamics that provide different values for alpha.
    • The accuracy of the model respect to the reactions curve must be calculated in a more simple way, just the SUM(abs(yp-ym)) without any segmentation for the identification points.
    • The method must be compared against other fractional order identification methods (2 or 3), as well as with the integer order ones (at least 2).

Once again I like the general idea very much, but I think it needs to be improved.

Reviewer 2 Report

The paper addresses a relevant problem in fractional modeling and control, that is, the identification of a fractional system by means of a simple procedure that can be easily used in industrial plants.

The paper is technically correct and the presence of experimental results makes it stronger. 

The main point I would better clarify is the advantage given by the use of a fractional model w.r.t. to an integer one (say, first-order-plus-dead-time model). In particular, by considering the experimental setup, what is the result if a method for integer-order systems is applied to the step response?

Is the improvement obtained by using a FFOPDT model significant? Is the additional effort to consider the fractional order model worth to obtaining a better model and, eventually, a better control performance? 

A discussion about this issue would significantly improve the paper.

Reviewer 3 Report

The article presents an identification procedure for fractional first-order plus dead-time (FFOPDT) models, which is based on the information obtained from three points on the process reaction curve. The procedure has been validated considering three symmetrical and asymmetrical points on the reaction curve. Simulation examples are given to verify the simplicity and effectiveness of this identification procedure. Also, a thermal process-based experimental setup has been used to confirm the applicability of the proposed method. To this reviewer, the article is technically well presented and clearly shows the effectiveness and performance of the proposed identification method. Only a minor revision of the English and the edition of the article is needed.

Round 2

Reviewer 1 Report

The reviewer thank the authors for the revision and the response to each comment.

All concerns are more-less addressed, even the reviewer point of view or perspective is different.

My only request is to emphasize which real processes include dynamics for the selected range of alpha [0.5, 1.1] (where the method will be used). Also, for easy reading to provide a resume about which is the final selection for the points to identify the model.

Finally, as suggestion it will be welcome to include as future work the possibility to extend the alpha range for larger values of alpha (including underdamped systems).
